# Learning to Decouple the Lights for 3D Face Texture Modeling

**Tianxin Huang**[1]      **Zhenyu Zhang**[2]      **Ying Tai**[2]      **Gim Hee Lee**[1]

[1]School of Computing, National University of Singapore
[2]Nanjing University
huangtx@nus.edu.sg, gimhee.lee@nus.edu.sg

## Abstract

Existing research has made impressive strides in reconstructing human facial shapes and textures from images with well-illuminated faces and minimal external occlusions. Nevertheless, it remains challenging to recover accurate facial textures from scenarios with complicated illumination affected by external occlusions, e.g. a face that is partially obscured by items such as a hat. Existing works based on the assumption of single and uniform illumination cannot correctly process these data. In this work, we introduce a novel approach to model 3D facial textures under such unnatural illumination. Instead of assuming single illumination, our framework learns to imitate the unnatural illumination as a composition of multiple separate light conditions combined with learned neural representations, named Light Decoupling. According to experiments on both single images and video sequences, we demonstrate the effectiveness of our approach in modeling facial textures under challenging illumination affected by occlusions. Our codes would be open sourced at https://github.com/Tianxinhuang/DeFace.git.

## 1 Introduction

Recently, 3D face reconstruction has made significant progress [4, 12, 13, 2, 30, 22] with the rapid development of digital human and meta-universe technologies. These techniques have become increasingly proficient in recovering details of face shapes and textures from single images or video sequences. Their performances have been particularly commendable on well-illuminated faces.

Despite the advancements, the real world presents more complex scenarios. As shown in the blue and red rectangled regions of input images in Fig. 1, self occlusions from facial parts such as the nose, or external occlusions such as hats or hairs introduce illumination changes and produce shadows in certain regions. Although recent works [16, 2, 21, 23, 22] replace the linear models [6, 4] with non-linear GAN-inversion-based textured models to greatly enhance the quality of textures and complete occluded facial areas, they still rely on the assumption that the environment illumination is single and uniform. As the illumination affected by self and external occlusions is unnatural, it may deviate drastically from the single and uniform illumination assumption, which would bring unexpected influence to the textures modelled with existing methods.

As shown in Fig. 1-a, methods [16, 10, 2] based on the diffuse-only texture and mesh render [10] bake both shadows caused by self occlusion of the nose and external occlusion of the hat onto the texture, where the re-rendered output also appears unrealistic and lacks authenticity. Recent works proposed by Dib *et al.* [12, 13, 11] combine local reflectance texture model incorporating diffuse, specular, and roughness albedos with ray-tracing render [26] to implicitly model influence of the self occlusion. As shown in Fig. 1-b, these methods can create more realistic rendered output and avoid the shadows of self occlusion on the recovered texture in the blue rectangle, while the influence from external occlusion in the red rectangle remains unresolved.

38th Conference on Neural Information Processing Systems (NeurIPS 2024).

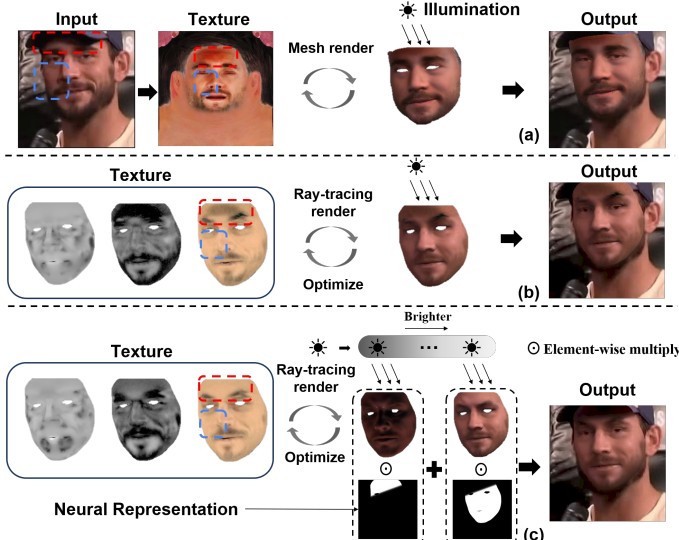

Figure 1: Blue and red rectangles mark regions affected by self and external occlusions, respectively. (a) Texture modeling with diffuse-only texture map. (b) Texture modeling based on diffuse, specular, and roughness albedos from local reflectance model [12], while optimizing with ray-tracing render. (c) Our method learns neural representations to decouple the original illumination into multiple light conditions, where the influence from external occlusions can be modeled as one of the conditions. White and black regions in the masks denote 1 and 0, respectively.

To mitigate the above-mentioned issues, we propose a face texture modeling framework for faces under complicated and unnatural illumination affected by external occlusions. Given the limited physical information about external occlusions presented in a single facial image or video sequence, accurately modeling how occlusion impacts illumination becomes nearly unattainable. Regarding this challenge, we propose to use *multiple separate light conditions* to imitate the local illumination of different facial areas after being affected by external occlusions. As shown in Fig. 1-c, our method can model the influence from external occlusion as one of the light conditions and eliminate its effect on the recovered texture as highlighted in the red rectangle, where the rendered output is also more realistic and closer to the input image especially on the forehead and eyes. Specifically, we use a spatial-temporal neural representation to predict masks for different light conditions, which are used to combine rendered results under multiple light conditions into final output. The number of light masks is adpatively adjusted during optimization. As external occlusions may directly cover parts of the face, we also introduce another neural representation to provide a continuous prediction for the available face region. Furthermore, our approach incorporates realistic constraints by introducing priors from the statistical model and pre-trained face perceptual models to ensure our extracted textures construct lifelike human faces.

Our contribution in this work can be summarized as:

- We present a new face texture modeling framework by learning to decouple the environmental illumination into multiple light conditions with neural representations;

- We propose realistic constraints to help improve the authenticity of texture by introducing human face priors;

- Extensive experiments on images and videos confirm that our method delivers more realistic renderings and improved facial textures compared to existing approaches under challenging illumination affected by occlusions.

## 2 Related Works

### 2.1 3D Morphable Model

The 3D Morphable Model (3DMM) [4, 15, 32, 6] is a widely employed linear statistical framework used for modeling the geometry and texture of faces. It is constructed based on aligned face images

using Principal Component Analysis (PCA). Within the 3DMM framework, the coordinates and colors of facial mesh vertices are computed through a linear combination of numerical parameters. These parameters can be estimated to enable face reconstruction through optimization-based fitting [4, 6] or learning-based approaches [10, 30, 43, 42], with the goal of minimizing the disparities between the rendered and original face images. Traditional 3DMMs, although effective, are limited by their linear PCA basis. Recent advancements [16, 21, 23, 44] have introduced non-linear basis by incorporating pre-trained mesh or texture decoder networks. In these methods, latent codes for these decoders are optimized to align with the original face images. This approach significantly enhances the robustness of the model when faced with issues such as occlusions and missing textures. However, such methods demand access to large, well-preprocessed datasets for effective pre-training.

## 2.2 Face Texture and Illumination Modeling

Texture and illumination modeling plays a pivotal role in 3D face reconstruction, directly impacting the color rendering in generated images. Accurate modeling for 3D face textures and environmental illumination is vital for subsequent applications such as face relighting [36, 18, 31] or animation [38, 30]. To achieve high-resolution rendering, recent approaches [9, 21, 16, 3] usually opt for UV mapping to model facial texture, as opposed to the earlier methods that focused on vertex colors [32]. The UV map can either be initialized with a PCA basis [15] or estimated through the use of a pre-trained non-linear texture decoder [22, 2, 16]. There are two commonly used models to estimate human skin reflectance for the texture modeling framework: the Lambertian model [1] and the Blinn-Phong model [5]. The Lambertian model is more computationally efficient, while the Blinn-Phong model produces more realistic rendering results by accounting for specular attributes in textures.

As capturing environmental illumination directly can be challenging, most existing methods assume it as primarily uniform Spherical Harmonics [33, 34]. This approach, however, tends to "bake" shadows cast by facial characters or external occlusions into the textures. Although method [45] introduce implicit texture modeling to achieve better face rendering performances, it requires multi-view images under uniform illumination for optimization, which is not appropriate for face images affected by occlusions. 2D shadow removal methods [41, 17, 28] try to eliminate the influence of shadows directly with networks. These methods have higher inference efficiency, while the size of their training set may greatly affect their performances. Notably, Dib *et al.* [12, 13, 11] have introduced techniques that implicitly model self-shadows through ray-tracing, demonstrating exceptional performance on faces with severe self occlusions. Despite these advancements, existing methods usually encounter challenges in handling shadows caused by external occlusions such as hats or hair, tending to take the shadows as part of textures.

To address these issues, we propose a novel framework to recover clear textures under challenging illumination affected by external occlusions. Our approach achieves this by decoupling the unnatural affected illumination into multiple light conditions using learned neural representations.

## 3 Our Methodology

### 3.1 Problem Definition

This work focus on the texture modeling problem of human faces under challenging environment illumination affected by external occlusions. Given an input image or video sequences $I_{in}$ taken from affected illuminations, our task is to recover clear and accurate texture $T$ from $I_{in}$ and ensure that $T$ can synthesize results $I_{out}$ close to $I_{in}$.

### 3.2 Overall Illustration

As illustrated in Fig. 2, instead of making the assumption that the face is exposed to a single and uniform illumination, we propose to model $n$ possible separate light conditions $\gamma_1 \sim \gamma_n$ in the environment, where the masks for possible regions $M_N$ are predicted with a neural representation $f(\cdot)$. Effective masks $M_L$ and rendered faces $I_{Rs}$ are selected from original $M_N$ and rendered $I_{Rn}$ during optimization with the Adaptive Condition Estimation (ACE) strategy. The final rendered face under multiple light conditions is $I_R = \sum I_{Rs} \odot M_L$, which is merged with the input images to construct the output $I_{out} = I_R \odot M_o + I_{in} \odot (1 - M_o)$. The face shape is modeled with statistical

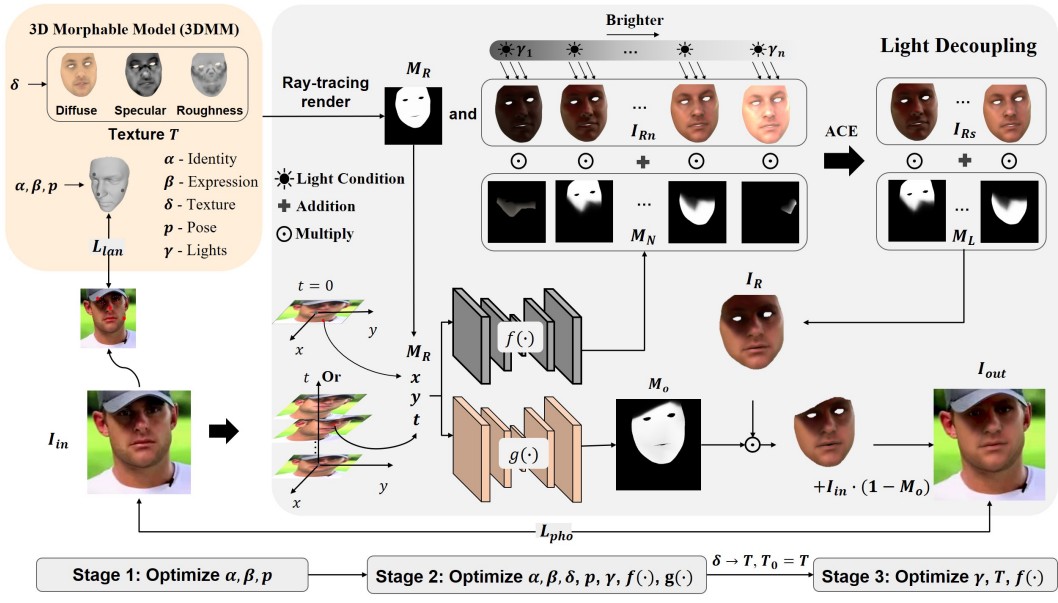

Figure 2: Illustration of our framework. The pipeline is proposed to recover texture $T$ and 3DMM statistical coefficients $\alpha, \beta, \delta, p, \gamma$ from the input image $I_{in}$. The statistical coefficient $\delta$ is used to initialized $T$. Render mask $M_R$ and Faces $I_{Rn}$ under $n$ light conditions $\gamma = \{\gamma_1 \sim \gamma_n\}$ are acquired through ray-tracing rendering. $f(\cdot)$ and $g(\cdot)$ are neural representations predicting light masks $M_N$ and facial region mask $M_o$. ACE is introduced to select effective masks $M_L$ and rendered faces $I_{Rs}$. $I_{Rs}$ are combined into $I_R$ with $M_L$, where $I_R$ is merged with surroundings in $I_{in}$ with $M_o$ to construct output image $I_{out}$. $L_{pho}$ and $L_{lan}$ are photometric loss and landmark loss, respectively.

coefficients $\alpha$, $\beta$, and $p$. Following [12], they are mainly optimized with landmark loss:

$$L_{lan} = \frac{1}{|q_{in}|} \|q_{out} - q_{in}\|_2, \tag{1}$$

where $q_{in}$ is 2D key points detected from $I_{in}$. $|q_{in}|$ is the number of points in $q_{in}$. $q_{out}$ denotes projected 3D key points on 2D plane. The color-related variables such as textures and lights are mainly optimized with photometric loss:

$$L_{pho} = \frac{1}{|I_{in}|} \|I_{out} - I_{in}\|, \tag{2}$$

where $|I_{in}|$ is defined as the number of pixels of input image $I_{in}$. More details are presented below.

### 3.3 Light Decoupling

**Light Condition Initialization.** Following [12, 13] using $B = 9$ bands Spherical Harmonics (SH) and ray-tracing rendering to model the illumination under self occlusions, we use $n$ separate SH to model $n$ possible light conditions. The coefficients are then simply initialized as $\gamma_i = 2 \cdot \frac{i}{n} \boldsymbol{I} - 1$ for $\gamma_i \in \gamma = \{\gamma_1, .., \gamma_n\}$, where $\boldsymbol{I}$ is a all-one matrix with the same shape as the SH coefficient. Please note that we use each SH to imitate the local illumination after being affected by the external occlusion, instead of the global natural illumination. This means that we do not need to consider physical influence of occlusion in each SH. Instead, we optimize each SH independently to directly imitate the illumination in different face regions.

**Neural Representations for Face Segment.** To decouple the illumination into multiple light conditions, we design a pair of spatial-temporal continuous neural representations to segment the face into regions for different light conditions. As illustrated in Fig. 2, spatial positions $x, y$ and temporal position $t$ of pixels are normalized and embedded into a coordinates system. $t$ is decided by the number of frames in the input image/video sequence. For the single image reconstruction, $t$ is set as a constant 0, where it would be $i/k$ for the $i_{th}$ frame of a $k$ frames video sequences. For the convenience of learning, $(x, y)$ are normalized into $[-1.0, 1.0]$ and $t$ is normalized into $[0.0, 1.0]$.

A Multi-Layer Perceptron (MLP) $f(\cdot)$ is then introduced to predict the probability of assignment to each light condition. Given the render mask $M_R$ in the ray-tracing-assisted rendering, the light masks can be obtained as $M_N = M_R \odot f(x, y, t)$. Effective masks $M_L$ are selected from $M_N$ with Adaptive Condition Estimation (ACE) to provide the final segment for different light conditions.

Similarly, we use another MLP $g(\cdot)$ to predict the probability that each pixel belongs to the face regions to avoid the influence from direct occlusion such as hat or hair. The Face mask is thus given by: $M_o = M_R \odot g(x, y, t)$. A pre-trained semantic segmentation network [27] is introduced for distillation to $g(\cdot)$, while the probability for labels of all face components such as eyes, mouth, or nose are added together to construct a classifier $h(\cdot)$ to predict the association of a pixel at $x, y, t$ to the face. The distillation loss $L_{seg}$ can be simply defined as:

$$L_{seg} = \frac{1}{|I_{in}|} \|g(x, y, t) - h(x, y, t)\|_2. \tag{3}$$

Note that $g(\cdot)$ is co-optimized together with the $L_{pho}$ and $L_{seg}$ to further distinguish some occlusions hard to be fitted with the 3DMM statistical model.

**Adaptive Condition Estimation (ACE).** As the complexity of illumination is mutative in different surroundings, we design a strategy to estimate the number of light conditions existing in this environment during optimization. Specifically, light masks with larger area than a pre-defined threshold $\epsilon$ in $M_N$ are preserved in $M_L$, while smaller ones are dropped with corresponding light conditions and not further optimized in later iterations. Given the number of light masks $M_N$ same as light condition $n$, then $M_N = \{M_N^i\}_{i=1}^n, M_L = \{M_L^i\}_{i=1}^{n_L} = \{M_N^i | \frac{1}{|I_{in}|} \sum_{m \in M_N^i} m > \epsilon, i <= n\}$. $n_L <= n$ is the number of preserved light conditions. $m$ denotes the pixel in $M_N^i$. We introduce an regularization $L_{area}$ to to encourage the area concentration on fewer masks:

$$L_{area} = \frac{1}{|I_{in}|} \sum \frac{1}{n} \sum_{i=1}^n e^{-\|M_N^i - \frac{\sum_{i=1}^n M_N^i}{n}\|_2} - 1, \tag{4}$$

where $|I_{in}|$ is the number of pixels in the mask.

To ensure each light condition is mainly contained in one mask of $M_L$, we introduce a binary regularization $L_{bin}$ to ensure that the predicted probability for each light condition tends to 0 or 1:

$$L_{bin} = \frac{1}{|I_{in}|} \sum \frac{1}{n_L} \sum_{i=1}^{n_L} e^{-\|M_L^i - \frac{\sum_{m \in M_L^i} m}{|I_{in}|}\|_2} - 1, \tag{5}$$

where $m \in M_L^i$ denotes each pixel value in $i^{\text{th}}$ light mask $M_L$. $n_L$ is the number of masks in $M_L$.

Note that $L_{area}$ is different from $L_{bin}$ as it pushes the mask values $M_N^i$ far from the mean of all masks, while $L_{bin}$ pushes mask values away from mean of different positions in the same mask $M_L^i$. ACE is executed at a specific iteration $iter_0$ to select $M_L$ from $M_N$. We use $L_{area}$ to encourage area concentration before $iter_0$, while using $L_{bin}$ to push the mask to 0 or 1 after $iter_0$.

## 3.4 Realistic Constraints

To ensure the reconstructed texture is reasonable and lifelike, we propose global prior constraint $L_{GP}$, local prior constraint $L_{LP}$, and human prior constraint $L_{HP}$ by introducing both priors from 3DMM statistical model and the pre-trained perceptual model.

**Global Prior Constraint.** The global prior constraint is used to ensure the consistency of overall hue between optimized texture $T$ and initialized texture $T^0$ from the 3DMM statistical model calculated with $\delta$. As the colors mainly come from diffuse albedo in $T$, we estimate the overall hue of $T^0$ with a K-means algorithm on its diffuse albedo $T_D^0$ and get a $4 \times 4$ color matrices $C$. Given the diffuse albedo of Texture $T$ as $T_D$, this term is given by:

$$L_{GP} = \frac{1}{|I_{in}|} \sum \min_{c \in C} \|T_D - c\|_2. \tag{6}$$

**Local Prior Constraint.** The local prior constraint is used to ensure the local smoothness of optimized texture $T$ by constraining its local variation to be similar to $T^0$. Let us define the diffuse, specular, and roughness albedos of texture $T$ as $T_D, T_S, T_R$, respectively. We set the local variations

---

**Algorithm 1** Training Process

---

1: Set the number of $iter_0 \sim iter_3$
2: **Stage 1:**
3: **for** $n = 1$ **to** $iter_1$ **do**
4:     Calculate the Loss of Stage 1 : $L_1 = L_{lan}$
5:     Update coefficients with $L_1$: $\nabla_{\alpha, \beta, p} L_1$.
6: **end for**
7: **Stage 2:**
8: **for** $n = 1$ **to** $iter_2$ **do**
9:     **if** $n < iter_0$ **then**
10:         Calculate the Loss of Stage 2 on $M_N$ and $I_{Rn}$:
11:         $L_2 = \omega_0 L_{pho} + \omega_1 L_{lan} + \omega_2 L_{seg} + \omega_3 L_{area}$
12:     **else if** $n = iter_0$ **then**
13:         Select $M_L$ and $I_{Rs}$ from $M_N$ and $I_{Rn}$ with ACE;
14:     **else**
15:         Calculate the Loss of Stage 2 on $M_L$ and $I_{Rs}$:
16:         $L_2 = \omega_0 L_{pho} + \omega_1 L_{lan} + \omega_2 L_{seg} + \omega_4 L_{bin}$
17:     **end if**
18:     Let $\theta_f$, $\theta_g$ be the parameters of $f(\cdot)$ and $g(\cdot)$.
19:     Update coefficients with $L_2$: $\nabla_{\alpha, \beta, p, \delta, \gamma, \theta_f, \theta_g} L_2$.
20: **end for**
21: **Stage 3:**
22: Extract texture $T$ from $\delta$ as a separate variable.
23: **for** $n = 1$ **to** $iter_3$ **do**
24:     Calculate the Loss of Stage 3 :
25:     $L_3 = \omega_0 L_{pho} + \omega_5 L_{GP} + \omega_6 L_{LP} + \omega_7 L_{HP}$
26:     Update coefficients with $L_3$: $\nabla_{T, \gamma, \theta_f} L_3$.
27: **end for**

---

of $T_D$, $T_S$, $T_R$ as $N_D$, $N_S$, $N_R$, which are computed with the $5 \times 5$ neighbors around each pixel, $N_D = T_D - \text{Neighbor}(T_D)$. $N_D^0$, $N_S^0$, $N_R^0$ are local variations calculated from $T^0$. The local prior constraint $L_{LP}$ is then defined as:

$$L_{LP} = \frac{1}{|I_{in}|}(\|N_D - N_D^0\| + \|N_S - N_S^0\| + \|N_R - N_R^0\|). \tag{7}$$

**Human Prior Constraint.**   Human prior constraint is introduced to enhance the optimization of texture $T$ in Stage 3 with a face recognition network FaceNet [35] pre-trained on large dataset such as VGGFace2 [7] or Casia-Webface [39]. The recognition model pre-trained on VGGFace2 and Casia-webface tends to classify the input images to 8,631 and 10,575 identities with different genders, ethnicity, *etc*. In this work, we propose a human prior constraint by maximizing the probability that the rendered face is recognized on one of the identities by FaceNet. Defining the FaceNet as $f_r(\cdot)$, the human prior constraint can then be written as:

$$L_{HP} = \sum \min_{s \in f_r(I_{Rs})} \|1 - s\|_2. \tag{8}$$

$I_{Rs}$ are rendered under multiple light conditions. $L_{HP}$ constrains the textures to be more reasonable.

### 3.5   Training Pipeline

The pipeline of the entire training process of our proposed framework is presented in Alg. 1. As illustrated in Fig. 2, the training of our proposed method consists of 3 stages, which is similar to NextFace [12]. In Stage 1, the expression $\beta$, shape $\alpha$, and pose $p$ coefficients are optimized to construct the basic face shape. In Stage 2, all coefficients, and the neural representations $f(\cdot)$ and $g(\cdot)$ are optimized together to reconstruct the face with statistical model. In Stage 3, the texture $T$ is directly optimized without coefficient $\delta$, while $\gamma$ and $f(\cdot)$ are optimized with small learning rates. $\omega_0 \sim \omega_7$ are pre-defined weights. We use Adam optimizer [19] for optimization.

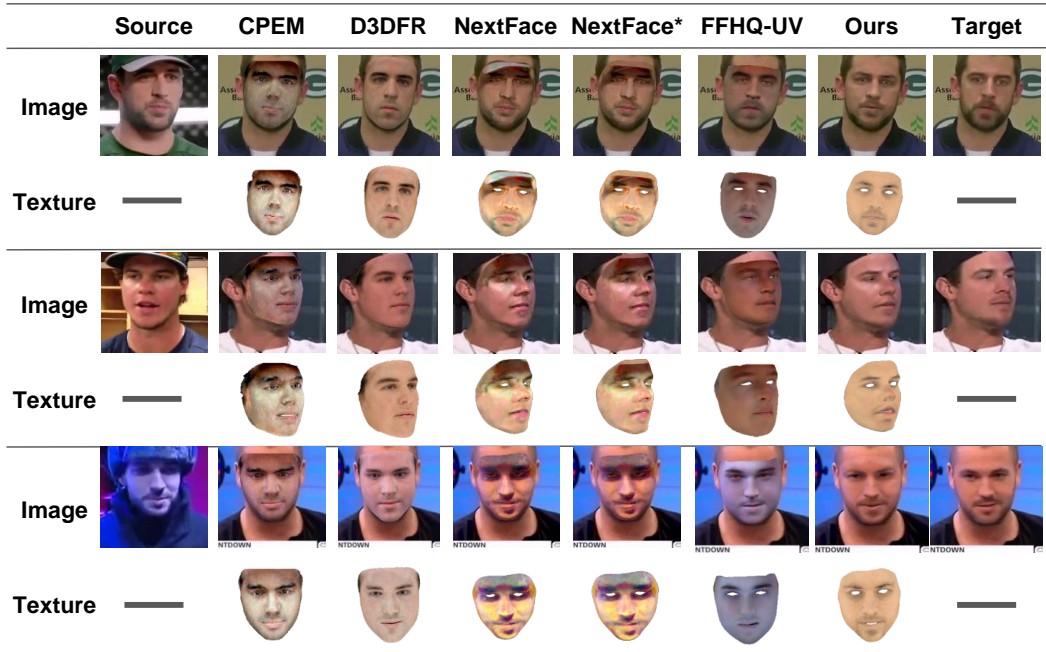

Figure 3: Comparison on Voxceleb2 images. The diffuse albedo is visualized as the texture because it contains most of the color information. Textures from source images are used to synthesize the target images. NextFace* denotes results optimized within regions selected with face parsing [27]. We do not have textures for CPEM [30] or D3DFR [10] as they predict vertex colors instead of uv textures.

## 4 Experiments

### 4.1 Dataset and Implementation Details

We follow NextFace [12] for the ray-tracing rendering and multi-stage organization of optimization pipeline. Detailed hyper-parameter settings can be found in our supplementary. For our experiments, we utilize two datasets: Voxceleb2 [8] and CelebAMask-HQ [29, 24]. VoxCeleb2 [8] is a diverse dataset encompassing numerous videos collected from interviews, movies and videos, where the same person may have multiple separate videos. CelebAMask-HQ [29, 24] is a large scale face image dataset with fine attributes annotation and high resolution, widely used in face editing and generation.

**Evaluation.** We construct a collection of evaluation data with challenging illumination affected by external occlusions for comparative analysis. This collection includes 38 pairs of single images, 24 pairs of videos from Voxceleb2 [8] with $256 \times 256$ resolution, and 62 single images from CelebAMask-HQ [29] with $512 \times 512$ resolution. Each pair consists of source images affected by external occlusions and target images without occlusion, both from the same person, where each video is sampled to 8 frames for sequence-based comparisons.

To quantitatively assess the quality of recovered textures, face textures extracted from the occluded source images are leveraged to synthesize the unoccluded target images. Specifically, we optimize source and target images separately following Sec. 3. Then, keeping the face shape, pose, and illumination invariant, we replace the target image's texture with the source's and re-render it to the synthetic result. This allows us to measure texture quality by quantifying the differences between the synthesized target images and the original target images. We also introduce differences between original and reconstructed source images within facial regions acquired by face parsing [27] as an assistant metric to see if textures restore source images well.

Our quantitative comparison employs three metrics: PSNR, SSIM, and perceptual error LPIPS [40] of AlexNet [20]. State-of-the-art methods D3DFR [10], CPEM [30], NextFace [12], and RGB Fitting methods proposed in FFHQ-UV [2] are introduced for comparison. For D3DFR, we adopt its enhanced PyTorch version. As NextFace [12] does not distinguish the face and external occlusion when optimizing the texture, we introduce **NextFace\*** in subsequent comparison by adding face parsing [27] to select the face region during optimization.

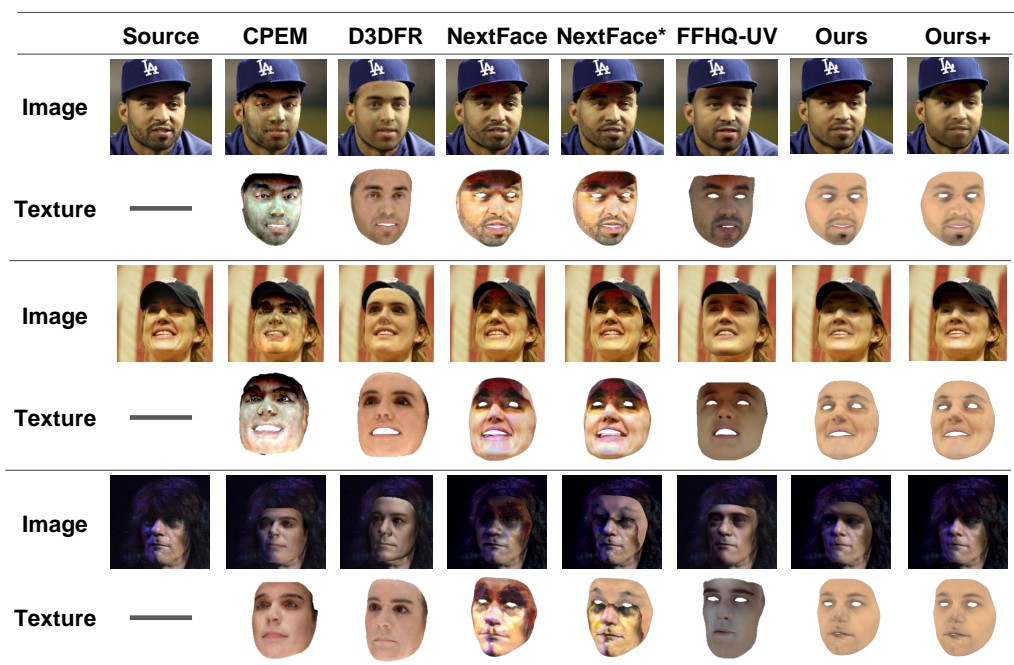

|  | Source | CPEM | D3DFR | NextFace | NextFace* | FFHQ-UV | Ours | Ours+ |
|---|---|---|---|---|---|---|---|---|
| **Image** | | | | | | | | |
| **Texture** | —— | | | | | | | |
| **Image** | | | | | | | | |
| **Texture** | —— | | | | | | | |
| **Image** | | | | | | | | |
| **Texture** | —— | | | | | | | |

Figure 4: Comparison results on the CelebAMask-HQ dataset. Ours and Ours+ denote our rendered results $I_R$ directly overlapped onto original images, and results combined with environments: $I_{out} = M_o \odot I_R + (1 - M_o) \odot I_{in}$, respectively.

Table 1: Quantitative comparison on single images from Voxceleb2. Source and Target denote differences evaluation on reconstructed source images and synthetic target images. LPIPS is multiplied with $10^2$. Underline and **bold** mark the suboptimal and optimal results, respectively.

|  |  | CPEM | D3DFR | NextFace | NextFace* | FFHQ-UV | Ours |
|---|---|---|---|---|---|---|---|
| Source | PSNR ↑ | 27.06 | 24.26 | **33.41** | 32.76 | 28.72 | 32.85 |
|  | SSIM ↑ | 0.87 | 0.87 | **0.95** | 0.94 | 0.92 | 0.94 |
|  | LPIPS ↓ | 10.22 | 11.25 | 7.37 | 7.88 | 8.32 | **6.40** |
| Target | PSNR ↑ | 24.70 | 27.23 | 23.74 | 24.21 | 25.03 | **29.22** |
|  | SSIM ↑ | 0.87 | 0.91 | 0.85 | 0.86 | 0.91 | **0.91** |
|  | LPIPS ↓ | 9.43 | 7.26 | 10.52 | 10.02 | 7.19 | **6.36** |

Table 2: Quantitative comparison on video sequences from Voxceleb2. Source and Target denote differences evaluation on reconstructed source sequences and synthetic target sequences.

|  |  | CPEM | D3DFR | NextFace | NextFace* | FFHQ-UV | Ours |
|---|---|---|---|---|---|---|---|
| Source | PSNR ↑ | 27.69 | 24.55 | 30.60 | 30.65 | 29.70 | **30.67** |
|  | SSIM ↑ | 0.87 | 0.87 | 0.92 | 0.92 | 0.92 | **0.92** |
|  | LPIPS ↓ | 9.21 | 9.92 | 8.17 | 8.45 | 7.64 | **7.62** |
| Target | PSNR ↑ | 24.33 | 26.20 | 24.15 | 24.32 | 24.35 | **29.15** |
|  | SSIM ↑ | 0.87 | 0.90 | 0.87 | 0.87 | 0.91 | **0.91** |
|  | LPIPS ↓ | 9.58 | 8.13 | 10.27 | 9.91 | 7.66 | **6.92** |

## 4.2 Evaluation on Voxceleb2

**Evaluation on Single Images.** We conduct an evaluation about the performance of our method in collected single images sourced from the Voxceleb2 dataset [8]. As shown in Fig. 3, our approach demonstrates a marked improvement in recovering clearer textures from the original images taken under challenging illumination affected by external occlusions. Our method also surprisingly recovers clear textures under strong colorful lights, which may benefit from the realistic constraints to keep textures reasonable. In contrast, existing methods tend to incorporate these occlusions, shadows, or colorful lights directly into the texture, resulting in less satisfactory outcomes.

Furthermore, our methods achieves superior results on the synthesis of target images and comparable performances on the reconstruction of source images as shown in Table 1. It confirms our method consistently recovers both accurate and clear textures. Although NextFace [12] performs a little better on PSNR and SSIM of source images, it performs the worst on synthetic target images as it actually severely over-fits source images. As shown in Fig. 3, it bakes shadows and occlusions to the textures, which confirms it is not appropriate for faces affected by external occlusions.

**Evaluation on Video Sequences.** We conduct an evaluation on video sequences collected from Voxceleb2 dataset [8] to further substantiate the efficacy of our approach. To adapt our framework

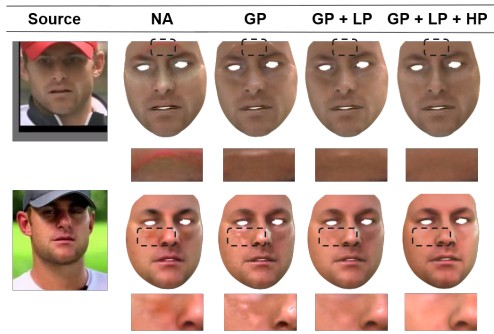

Figure 5: Ablation study for losses. GP, LP, and HP denote $L_{GP}$, $L_{LP}$, $L_{HP}$, respectively, while NA means to remove all of them.

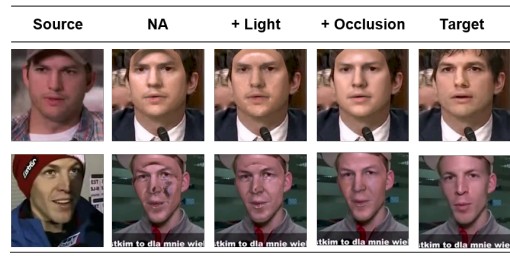

Figure 6: Ablation study for the neural representations. NA means to remove both $f(\cdot)$ and $g(\cdot)$, while **+ Light** and **+ Occlusion** denote adding $f(\cdot)$ and $g(\cdot)$, respectively.

Table 3: Quantitative ablation study for proposed Losses GP, LP, and HP.

| Metrics | NA | GP | GP+LP | GP+LP+HP (ours) |
|---|---|---|---|---|
| PSNR ↑ | 27.20 | 28.81 | 28.82 | **29.22** |
| SSIM ↑ | 0.88 | 0.89 | 0.90 | **0.91** |
| LPIPS ↓ | 8.34 | 6.78 | 6.49 | **6.36** |

Table 4: Quantitative ablation study for $f(\cdot)$, $g(\cdot)$.

| Metrics | NA | + Light ($f(\cdot)$) | + Occlusion ($g(\cdot)$) |
|---|---|---|---|
| PSNR ↑ | 25.19 | 27.52 | **29.22** |
| SSIM ↑ | 0.87 | 0.89 | **0.91** |
| LPIPS ↓ | 9.16 | 7.75 | **6.36** |

to video sequences, we share texture, illumination and shape coefficients across all frames during optimization. As D3DFR [10] and FFHQ-UV [2] do not provide support for sequences, we recurrently apply them to each single image. Quantitative comparison in Table 2 demonstrates that our method constantly performs superior to existing methods on texture modeling from continuous video sequences. Please check our Supplementary for corresponding qualitative results.

### 4.3 Evaluation on CelebAMask-HQ

We extend our comparisons on CelebAMask-HQ [29]. As this dataset lacks multiple images from the same identity, we focus on evaluating the performance based on the recovered textures and the reconstructed results of source images. The outcomes of our assessment are visually depicted in Fig. 4. We observe that our method continues to perform well in the task of recovering clear textures from challenging input. Additionally, our reconstructed results exhibit a higher degree of realism on the reconstructed results when compared to the outcomes produced by other methods.

### 4.4 Evaluation on Images with Diverse Shadows [41]

To validate the performances of our method more sufficiently, we further evaluate our method on the single image dataset proposed by [41], which includes 100 images affected by manually created external shadows, as well as corresponding ground truths. The quantitative and qualitative comparisons are presented in Table 5 and Fig. 7. We can observe that our method still outperforms other methods under faces with diverse shadows.

Table 5: Quantitative comparisons on images with diverse shadows.

| Metrics | CPEM | D3DFR | NextFace | NextFace* | FFHQ-UV | Ours |
|---|---|---|---|---|---|---|
| PSNR ↑ | 24.75 | 23.02 | 32.10 | 31.92 | 26.93 | **32.13** |
| SSIM ↑ | 0.83 | 0.83 | 0.94 | 0.94 | 0.89 | **0.94** |
| LPIPS ↓ | 10.99 | 12.76 | **5.26** | 5.34 | 8.71 | 6.29 |
| PSNR ↑ | 23.34 | 25.02 | 21.83 | 21.93 | 24.00 | **28.97** |
| SSIM ↑ | 0.84 | 0.87 | 0.84 | 0.84 | 0.88 | **0.92** |
| LPIPS ↓ | 9.31 | 9.11 | 9.48 | 9.69 | 7.82 | **7.00** |

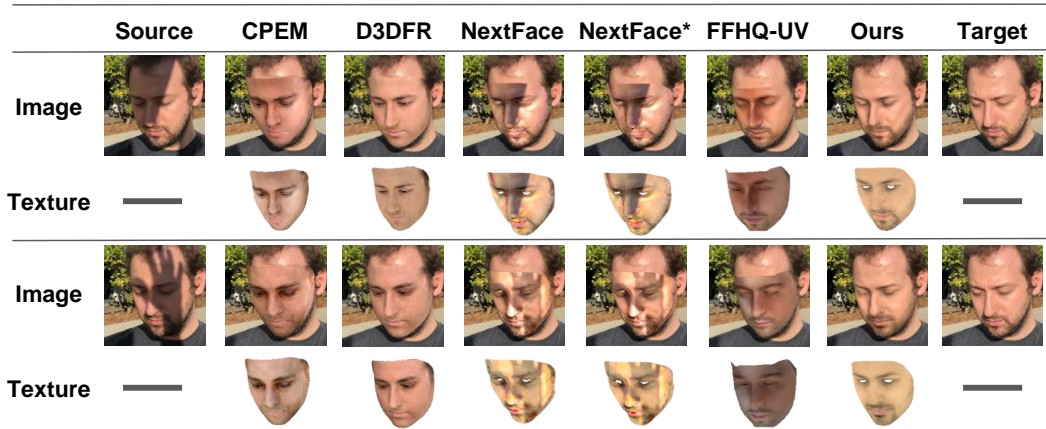

Figure 7: Comparison on images with diverse shadows [41]. As [41] provides corresponding ground truths of shadow affected images, we directly use such ground truths as the target images.

### 4.5 Ablation Study

**Ablation Study on Losses.** In this section, we analyze the impact of the proposed losses: $L_{GP}$, $L_{LP}$, and $L_{HP}$ by comparing the rendered results under relative brighter light conditions. As illustrated in Fig. 5, it becomes evident that the inclusion of the $L_{GP}$ loss plays a significant role in eliminating the abnormal colors on the texture, where $L_{LP}$ loss effectively remove large artifacts obviously different from human faces. $L_{HP}$ can further reduce slight unreasonable defects with priors from the pre-trained recognition model [35]. Quantitative comparisons in Table 3 also confirm that each proposed loss contributes to the final performances.

**Ablation Study on Neural Representations.** To ascertain the significance of the neural representations employed, we perform an analysis where we eliminate both the neural representations: $f(\cdot)$ and $g(\cdot)$ from the light decoupling pipeline. As depicted in Fig. 6, the neural representation for the decoupling of light conditions $f(\cdot)$ notably contributes in the removal of shadows artifacts from the results. Furthermore, the neural representation $g(\cdot)$ displays its efficacy in minimizing existing irregularities in the synthetic results, which achieves smoother and cleaner results. We also present corresponding quantitative comparisons in Table 4, which demonstrates that both $f(\cdot)$ and $g(\cdot)$ have obvious influence on the final performances.

## 5 Conclusion

In this paper, we present a novel framework dedicated to recover clear facial textures from images taken under challenging illumination affected by external occlusion. Our approach makes use of neural representations to decouple the original illumination into multiple separate light conditions across various facial regions. Then the affected complicated illumination can be modelled with the combination of different light conditions. Furthermore, we introduce well-established human face priors through the realistic constraints to enhance the realism of our results. According to experiments on single images and video sequences, our method consistently surpasses existing techniques to recover clearer and more accurate textures from faces under affected unnatural illumination.

## Acknowledgement

This research / project is supported by the National Research Foundation (NRF) Singapore, under its NRF-Investigatorship Programme (Award ID. NRF-NRFI09-0008).

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

# A  Appendix / Supplementary Material

## A.1  Limitation

We use AlbedoMM [37] to initialize the face texture and optimize neural representations. Limited by the capacity of AlbedoMM, the initialized face texture may not be accurate enough for subsequent optimization in some occasions. We can explore to resolve this problem by combining our method with non-linear texture modeling methods such as [22, 2] in our future work. Furthermore, the optimization of multiple light conditions makes our method slower than other single illumination methods [12, 2]. It costs 340s for a $256 \times 256$ image on 2080ti, which is slower than 240s of FFHQ-UV [2] but still affordable.

## A.2  Details of Parameter Settings

In Table 6, we present details about the hyper-parameters mentioned in Sec. 3. Although it seems that there are multiple parameters, *the setting is robust for both images and video sequences from adopted datasets*. We conduct all experiments on a Nvidia 2080ti GPU with a 2.9Ghz i5-9400 CPU. It costs 340s for a $256 \times 256$ image.

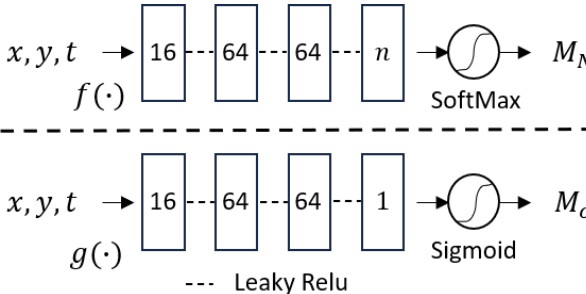

Figure 8: Detailed design of $f(\cdot)$ and $g(\cdot)$.

Table 6: Hyper-parameter settings. $n$ is the number of initial light conditions presented in Fig. 2.

|  | Parameter |
| --- | --- |
| $w_1 \sim w_7$ | 2e3, 1e-3, 1.5e2, 0.5, 25, 2e3, 2.0, 1.0 |
| Landmark | Mediapipe |
| $iter_0 \sim iter_3$ | 100, 2000, 400, 200 |
| $\epsilon$ | 0.17 |
| $n$ | 5 |

## A.3  Comparisons against the combination of 2D Shadow Removal and 3D Texture Modeling

Although the former mentioned existing 3D face texture modeling methods cannot process facial shadows from external occlusions directly, there are some 2D shadow removal methods [41, 17, 28] trying to eliminate the shadows directly from the images. Therefore, another alternative simple baseline is to pre-process the image with 2D shadow-removal networks before texture modeling. In this section, we introduce the most recent method [17] with a pre-trained model to pre-process the images before feeding them to compared methods mentioned in Sec. 4.1. The quantitative and qualitative comparison are presented in Table 7, Table 8 and Fig. 11, respectively.

Table 7: Comparisons against baselines with 2D shadow-removal pre-processing on [41].

|  | CPEM | D3DFR | NextFace | NextFace* | FFHQ-UV | Ours |
| --- | --- | --- | --- | --- | --- | --- |
| PSNR ↑ | 25.56 | 25.60 | 25.24 | 25.47 | 26.19 | **28.97** |
| SSIM ↑ | 0.86 | 0.88 | 0.87 | 0.87 | 0.90 | **0.92** |
| LPIPS ↓ | 8.56 | 8.68 | **6.37** | 6.41 | 7.02 | 7.00 |

Pre-processing with 2D shadow removal methods indeed improves the performances of baselines, while our method still outperforms them. From qualitative results, we observe that the shadow

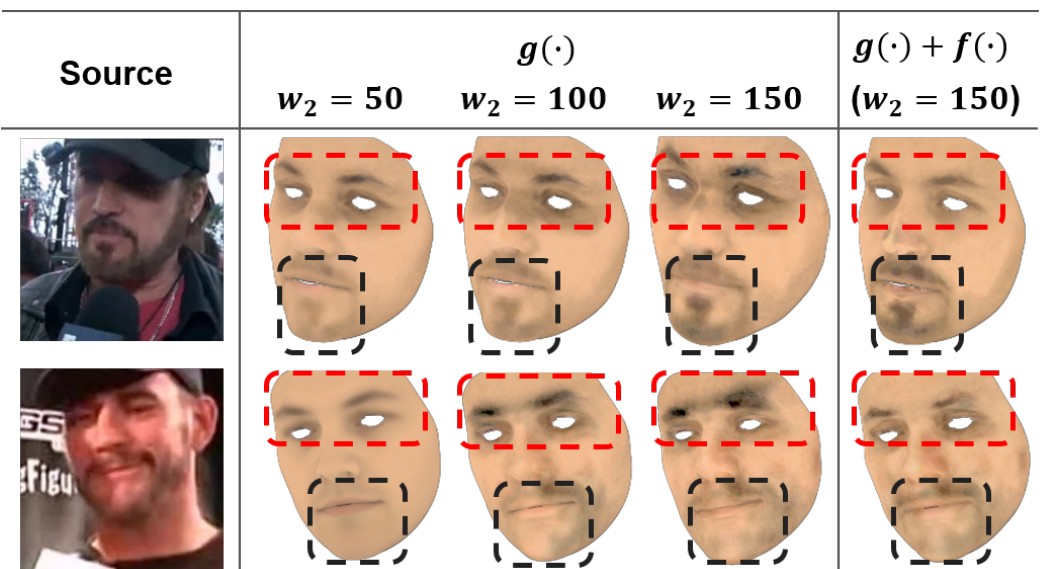

Figure 9: Discussion about the effect of $g(\cdot)$. $w_2$ is the weight to constrain $g(\cdot)$, defined in Alg. 1. The red and **black** rectangles mark shadow-affected regions and detailed textures, respectively. $g(\cdot)$ will weaken both shadows and details from textures when reducing $w_2$ to loose its constraint.

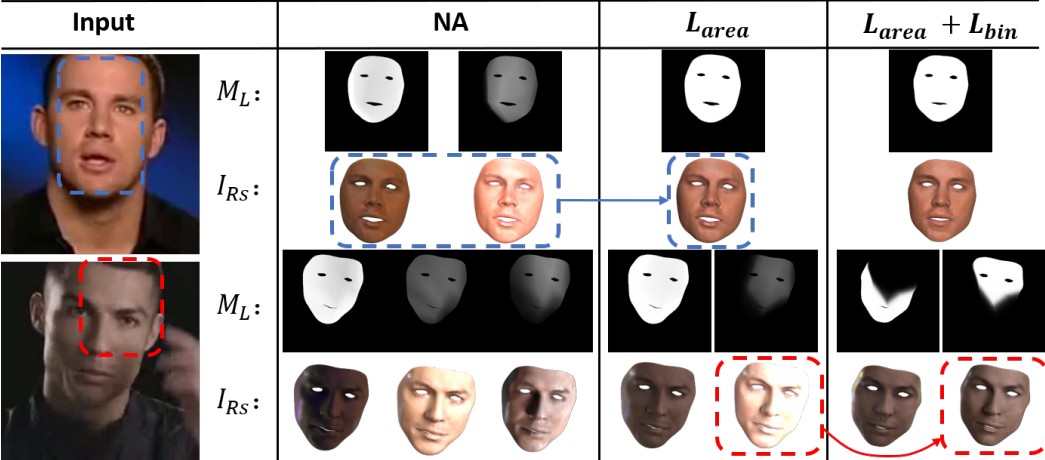

Figure 10: Ablation study for $L_{area}$ and $L_{bin}$ in ACE. NA denotes removing both $L_{area}$ and $L_{bin}$. $L_{area}$ can remove redundant light conditions as shown by the blue rectangle, while $L_{bin}$ ensures the light condition shown in the red rectangle region is consistent as our observation of the input. $M_L$ and $I_{Rs}$ are predicted masks and rendered faces defined in Fig. 2, respectively.

removal model cannot fully remove the external shadows in cases where the external shadows cover relatively large regions. Although modifying the shadow removal model to be more powerful may further improve performances, it goes beyond the range of this work. We can explore it in the future.

## A.4 Comparisons with Deocclusion methods

Besides the former mentioned shadow removal baselines, learning-based deocclusion methods [25, 14] can remove external occlusions from faces by predicting the occluded regions. Such operations also have the potential to deal with external shadows by directly treating the shadow regions as occlusions. In this section, we conduct a brief comparison with the most recent open sourced deocclusion method [25]. The quanlitative and qualitative results are presented in Table 9 and Fig. 12, respectively. The metrics are evaluated on the target images mentioned in Sec. 4.1. We can observe that the method [25] lose some facial details such as beards. The reason is that the deocclusion method [25] actually divides occlusion regions by the distances between input images

Table 8: Comparisons against baselines with 2D shadow-removal pre-processing on images from Voxceleb2.

|  | CPEM | D3DFR | NextFace | NextFace* | FFHQ-UV | Ours |
|---|---|---|---|---|---|---|
| PSNR ↑ | 24.84 | 26.78 | 23.77 | 24.51 | 25.35 | 29.22 |
| SSIM ↑ | 0.87 | 0.90 | 0.85 | 0.86 | 0.91 | 0.91 |
| LPIPS ↓ | 10.20 | 7.93 | 10.47 | 9.64 | 7.62 | 6.36 |

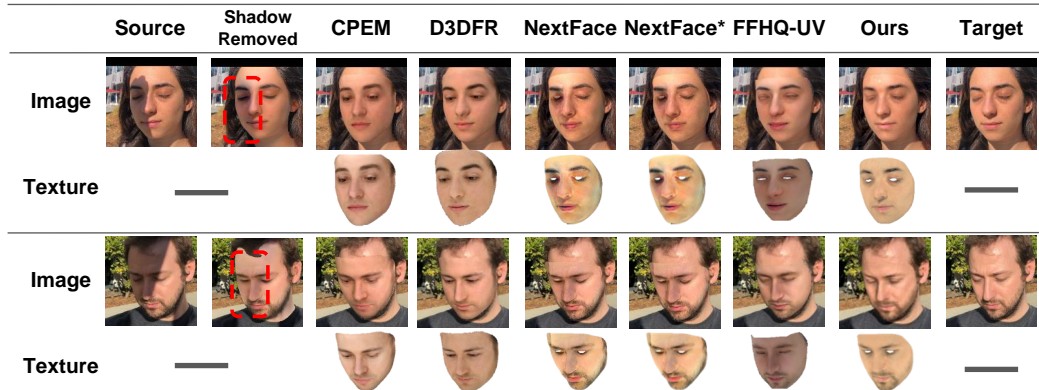

Figure 11: Qualitative Comparisons against baselines with 2D shadow-removal pre-processing.

and the reconstructed results from 3D Morphable Model (3DMM), which will also regard details hard to describe with 3DMM as occlusions. Our methods treat the shadows as different illumination instead of occlusions, which can preserve facial details in these regions in subsequent optimization.

### A.5    Discussion about the number of Sphere Harmonics (SH) bands.

In this work, we follow NextFace [1] to use 9-bands SH to model the local illumination, which can actually capture quite fine details. However, SH with more bands would have stronger ability for the modeling of illumination. To verify if the external shadows can be directly modelled with more bands SH, we also provide the quantitative results of our method modelled with only one single global SH in 9, 12, 15, 18 bands in Table 10, where we remove our light decoupling framework. We observe that increasing the number of bands in a single global SH yields quite limited improvements. A possible reason is that the external occluded shadows on human faces represent drastic illumination changes in relatively small areas, which may not be appropriately modeled as a single global SH during optimization.

### A.6    Discussion about the number of lighting conditions $n$.

In this work, we initialize $n$ different lighting conditions to imitate the illumination affected by external occlusion at the beginning, where some of them would be removed by ACE in subsequent processing. To explore the influence of different numbers of $n$, we conduct a ablation study for the number of $n$ used for initialization of lighting conditions in Table 11. Single images from VoxCeleb2 are used for evaluation. We observe that $n = 3$ produces sub-optimal results, likely because the number of lighting condition candidates is insufficient to model images with complex illuminations. In contrast, $n = 5, 7, 9$ yield good and similar results as there are enough initial lighting conditions, and any redundant ones are removed. The result for $n = 5$ is slightly better. While introducing larger $n$ and further adjusting the hyper-parameters might improve performance, it would also increase the optimization burden.

### A.7    Differences between Human Prior Constraint and Perceptual Loss

Please note that the rendered faces used to calculate Human Prior Constraint (HP) are calculated with $I_{Rs}$. As illustrated in Fig.2, $I_{Rs}$ includes rendered faces under multiple different lighting conditions.

Table 9: Comparisons against baselines with Deocclusion methods on Voxceleb2.

| Methods | Single Images | | Sequences | |
|---|---|---|---|---|
| | Deocclu | Ours | Deocclu | Ours |
| PSNR ↑ | 25.13 | **29.22** | 25.03 | **29.15** |
| SSIM ↑ | 0.88 | **0.91** | 0.88 | **0.91** |
| LPIPS ↓ | 14.09 | **6.36** | 14.18 | **6.92** |

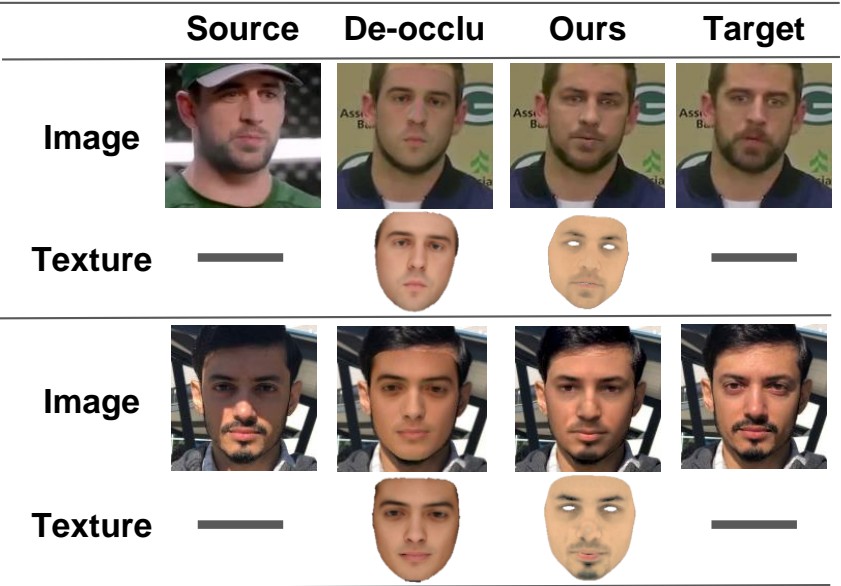

Figure 12: Qualitative Comparisons with the Deocclusion method [25].

We cannot use perceptual loss here because we do not have **corresponding ground truth images under the multiple decoupled lighting conditions**, where HP does not need such ground truths. Nonetheless, the perceptual loss can be indeed applied between the final rendered result $I_{out}$ and input image $I_{in}$. We present a comparison between such perceptual loss implementation and the HP in Table 12. We can see that HP still has better performances, which can confirm it provides more effective constraints for the textures through rendered faces under multiple lighting conditions.

### A.8 Discussion about $g(\cdot)$.

Both $f(\cdot)$ and $g(\cdot)$ have impacts to remove artifacts on the recovered textures. To confirm that they have principled differences, we implement a discussion about the effect of $g(\cdot)$. As presented in the third column of Fig. 9, $g(\cdot)$ cannot fully avoid the influence of external shadows on the faces. As presented in Sec. 3.3, $L_{seg}$ is introduced to constrain $g(\cdot)$ to ensure that $g(\cdot)$ predicts relatively accurate face regions. $L_{seg}$ is weighted by $w_2$ in optimization as defined in Alg. 3. Although the shadow effect can decrease when we reduce $w_2$ to allow $g(\cdot)$ to filter out more face regions, the detailed textures are equally weaken and may be fully removed as shown in the last row of Fig. 9. It confirms that $f(\cdot)$ is still essential for this framework.

### A.9 Analysis about $L_{bin}$ and $L_{area}$ in ACE.

In ACE mentioned in Sec 3.3, two regularization constraints $L_{area}$ and $L_{bin}$ are introduced. As the blue rectangle input face of Fig. 10 is not affected by any external occlusion, its illumination can be modelled with only one light condition. $L_{area}$ helps remove redundant light conditions. However, only using $L_{area}$ may create decoupled light conditions obviously different from the observation of input image, as illustrated in the red rectangle regions of Fig. 10. Adding $L_{bin}$ can ensure the

Table 10: Ablation study for the number of SH bands.

| B | 9 | 12 | 15 | 18 | Ours |
|---|---|---|---|---|---|
| PSNR ↑ | 25.87 | 25.26 | 25.27 | 25.34 | **29.22** |
| SSIM ↑ | 0.87 | 0.87 | 0.87 | 0.87 | **0.91** |
| LPIPS ↓ | 9.16 | 9.23 | 9.22 | 9.10 | **6.36** |

Table 11: Ablation study for the number of lighting conditions $n$.

| n | 3 | 5 (we use) | 7 | 9 |
|---|---|---|---|---|
| PSNR ↑ | 28.49 | **29.22** | 29.14 | 29.16 |
| SSIM ↑ | 0.90 | **0.91** | 0.91 | 0.91 |
| LPIPS ↓ | 6.37 | **6.36** | 6.37 | 6.39 |

predicted masks $M_L$ to be nearly binarized, which makes the decoupled light conditions consistent as the human observation of the input image.

### A.10 Differences between Stage 2 and Stage 3.

Illustrated in Fig. 2, our approach composes of both Stage 2 and Stage 3 to optimize the face texture. In this section, we conduct comparisons to demonstrate differences between textures from Stage 2 and Stage 3. The results are presented in Fig. 13. We can observe that textures from AlbedoMM [37]

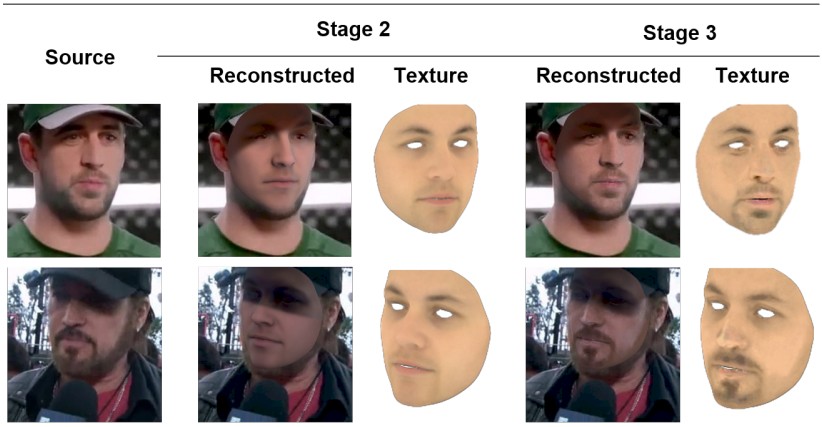

Figure 13: Differences between Stage 2 and Stage 3. In Stage 3, the texture is refined with details from the source image, such as the beard, to render a more realistic reconstructed image.

in Stage 2 are quite over-smoothed and lack of details. With Stage 3, details in source images are added to textures, which can reconstruct more realistic results than Stage 2.

### A.11 Visualization about $M_o$

We present an ablation study to confirm the effect of $g(\cdot)$ against direct segmented mask from face parsing [27] to predict $M_o$. The circled parts of Fig. 14 show that the parsed masks may be inappropriate due to the limitation of generalizability. We can see in the first row of Fig. 14 that the rendered result $I_R$ may have rough edges and artifact colors from the occlusion, while our method can avoid this problem by refining the parsed mask with $g(\cdot)$. Moreover, some parts may be missing on the parsed mask. As shown in the second and third rows of Fig. 14, artifacts show up in these unconstrained regions. Our method can complete these missing regions for more reasonable results.

Table 12: Comparison between HP and Perceptual loss.

| n | Perceptual Loss [10] | HP (ours) |
|---|---|---|
| PSNR ↑ | 28.72 | **29.22** |
| SSIM ↑ | 0.90 | **0.91** |
| LPIPS ↓ | 6.46 | **6.36** |

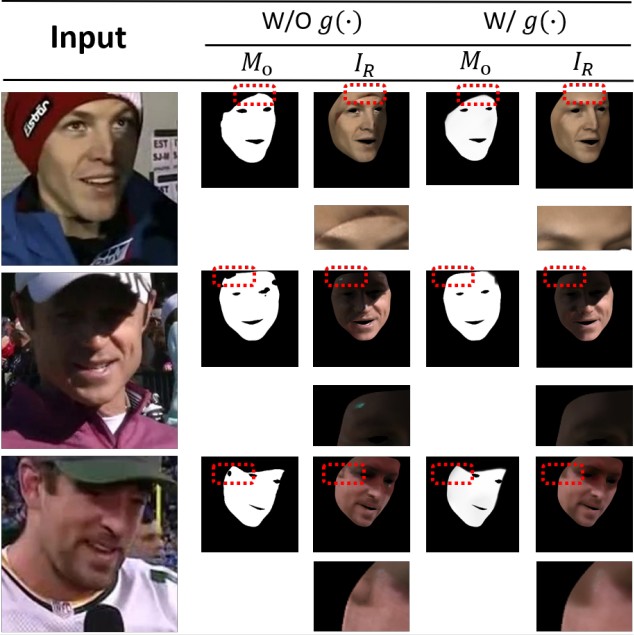

Figure 14: Ablation study for the usage of neural representation $g(\cdot)$. $W/O$ $g(\cdot)$ and $W/$ $g(\cdot)$ denote using parsed mask [27] and using $g(\cdot)$, respectively. $M_o$ and $I_R$ are the mask and rendered results, as mentioned in Sec. 3.

## A.12 Discussion about the failure cases.

Except the efficiency problem mentioned in Sec. A.1, our primary limitation is the initialization with AlbedoMM. As shown in Fig. 15, for faces with many high-frequency details, such as wrinkles, our method may lose these details during reconstruction. Replacing AlbedoMM with more powerful face representations could address this issue. We plan to explore this further in future work.

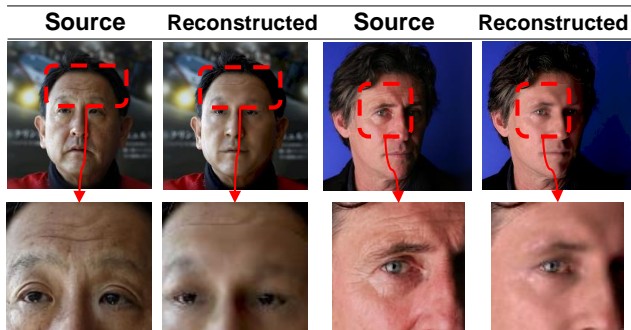

Figure 15: Some failure cases.

## A.13 Effect of Adaptive Condition Estimation

As described in Sec. 3.3, the Adaptive Condition Estimation (ACE) is proposed to select effective $M_L$ and $I_{Rs}$ from the initialized $M_N$ and $I_{Rn}$. To remove ACE, we use initialized $M_N$ and $I_{Rn}$ as $M_L$ and $I_{Rs}$ directly. As shown in Fig. 16, we can see that the rendered results $I_{out}$ are almost

| Input | W/O ACE | | | W/ ACE | |
|---|---|---|---|---|---|
| $I_{in}$ | $M_L(up)$ and $I_{Rs}(down)$ | | $I_{out}$ | $M_L(up)$ and $I_{Rs}(down)$ | $I_{out}$ |

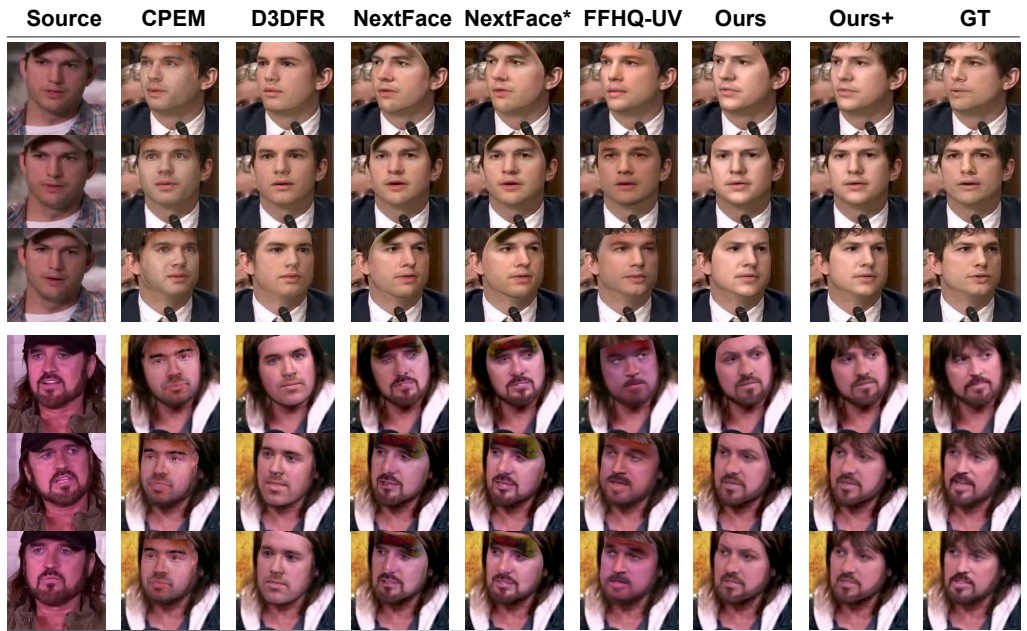

Figure 16: Ablation study for ACE. $W/O$ ACE and $W/$ ACE denote removing ACE by using $M_N$ and $I_{Rn}$ as $M_L$ and $I_{Rs}$, and using ACE to select $M_L$ and $I_{Rs}$, respectively.

Figure 17: Comparison results on the video sequences from Voxceleb2. Ours and Ours+ denote our rendered results $I_R$ directly overlapped onto original images, and results combined with environments: $I_{out} = M_o \odot I_R + (1 - M_o) \odot I_{in}$, respectively. The symbols are defined following Sec. 3.

the same, while the results optimized without ACE contain multiple redundant and inaccurate light conditions in $M_L$. It confirms that ACE can help remove these unnecessary light conditions and keep effective ones. With ACE, our method can decouple the original illumination affected by occlusions into light conditions more consistent with actual observations of input images.

## A.14 More Visualized Results

In this section, we present three representative examples from each sequence in Fig. 17. It is evident that our method performs the best in generating synthetic results close to the target sequences, exhibiting a high degree of realism. In contrast, other methods still produce less convincing outcomes due to negative effects from external occlusions. More results on images sourced from Voxceleb2 [8] and CelebAMask-HQ [24] are presented in Fig. 18 and Fig. 19. Our method still performs better. Please refer to the attached video for more results on sequences from Voxceleb2 [8].

| Source | CPEM | D3DFR | NextFace | NextFace* | FFHQ-UV | Ours | Target |
|--------|------|-------|----------|-----------|---------|------|--------|

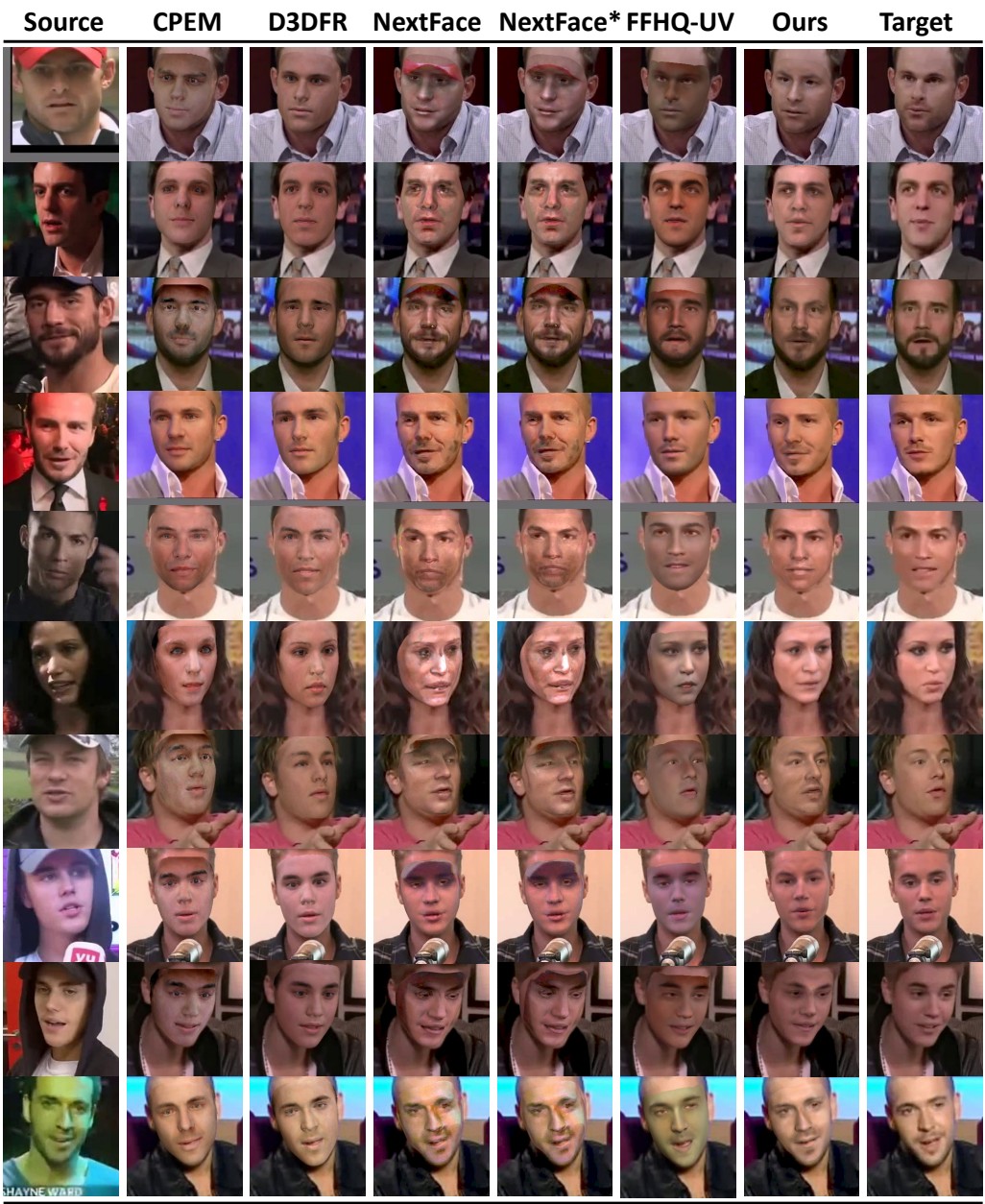

Figure 18: More results on single images from Voxceleb2 [8]. We can see that our method can synthesize more accurate target images based on textures from the source images, which validate the quality of our acquired textures.

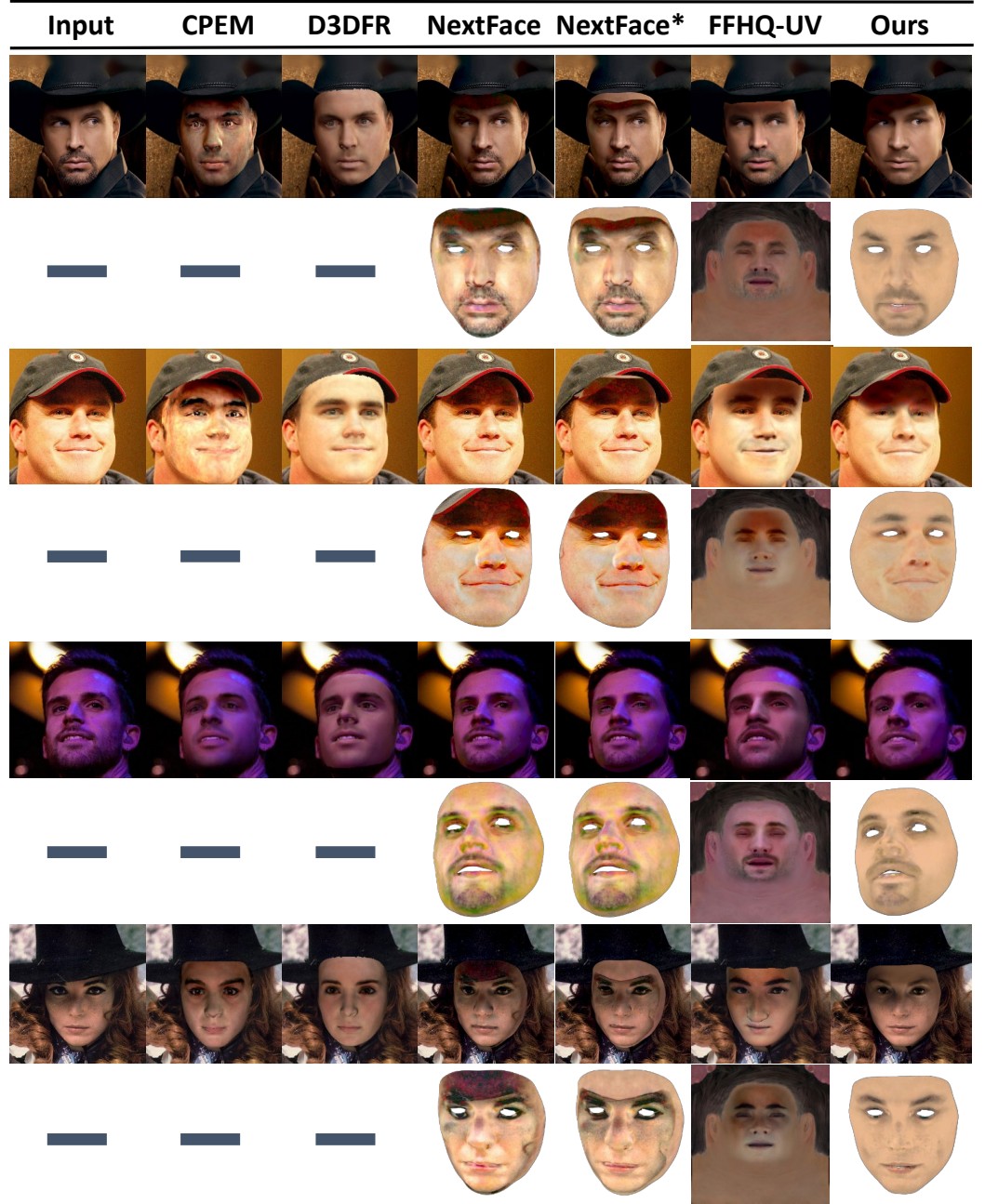

Figure 19: More reconstructed images/textures results on CelebAMask-HQ [24]. We can see that our method can reconstruct more realistic results, with clear textures without shadow effects.

