# OpenReview forum: "Learning to Decouple the Lights for 3D Face Texture Modeling"
_NeurIPS.cc/2024/Conference — NeurIPS 2024 poster_

### Official Review · Reviewer_NbDK · 2024-06-15

**Soundness:** 3
**Presentation:** 3
**Contribution:** 3
**Rating:** 7
**Confidence:** 4

**Summary:**

The paper presents a new face texture modeling framework by learning to decouple environmental illumination into multiple light conditions with neural representations. The proposed method can reconstruct a cleaner texture compared to previous work that use a single environment map. The proposed method is neat and reasonable. The paper is well-written.

**Strengths:**

* The first method to tackle the challenging but important problem of face reconstruction which is neglected by most existing works.
* The method is novel and neat. After reading the introduction and going through Fig.1, I believe the proposed method can well solve the problem.
* The paper is well-written and easy to follow.

**Weaknesses:**

* Line 119 says the n lighting coefficients are initialized differently. Can you provide some insights behind these design choices? Is the method sensitive to the initialization of lighting?

* In the proposed Human Prior Constraint, the rendered face is encouraged to be similar to a specific face in the FaceNet. Why not use the common perceptual loss as D3DFR?

* The texture of D3DFR in Figure 3 is missing. I know it reconstructs the BFM texture, which is not a diffuse albedo map. But I think it should also be provided. In addition, the FFHQ-UV's texture is presented in UV space while NextFace and the proposed methods' texture are presented in image space, I suggest presenting them all in UV space or all in image space.

**Questions:**

See Weaknesses.

**Limitations:**

The limitations are discussed.

---

> ### Author Rebuttal · Authors · 2024-08-06
>
> **Q1: Line 119 says the n lighting coefficients are initialized differently. Can you provide some insights behind these design choices? Is the method sensitive to the initialization of lighting?**
>
> **A1:** Sure. The lighting coefficients are just simply initialized as a uniform distribution between -1 to 1 as illustrated in Line 119, Sec. 3.3 of the paper. Such operation can provide $n$ initialized local illuminations ranging from dark to light. During the later optimization of Stage 2, the $n$ groups of lighting coefficients are optimized towards actual illumination. Coefficients that deviate significantly from the existing illumination in the images will have masks with smaller areas, as shown in the visualized $M_N$ of Fig.2 of the main paper. These coefficients with smaller masks are deemed less important and are removed according to a predefined threshold $\epsilon$ in ACE.
>
> Here, we also provide a ablation study for the number of $n$ used for initialization of lighting conditions below. Single images from VoxCeleb2 are used for evaluation.
>
> | n | 3 | 5 (we use) | 7  | 9 |
> |--|--|--|--|--|
> |PSNR $\uparrow$ | 28.49 |29.22 |29.14 | 29.16|
> |SSIM $\uparrow$ | 0.90 | 0.91 |0.91 | 0.91|
> |LPIPS $\downarrow$ | 6.37 | 6.36 | 6.37 | 6.39 |
>
> We observe that $n=3$ produces sub-optimal results, likely because the number of lighting condition candidates is insufficient to model images with complex illuminations. In contrast, $n=3, 5, 9$ yield good and similar results as there are enough initial lighting conditions, and any redundant ones are removed. The result for $n=5$ is slightly better. While introducing larger $n$ and further adjusting the hyper-parameters might improve performance, it would also increase the optimization burden. Therefore, we use $n=5$ in this work.
>
> **Q2: In the proposed Human Prior Constraint, the rendered face is encouraged to be similar to a specific face in the FaceNet. Why not use the common perceptual loss as D3DFR?**
>
> **A2:** Please note that the rendered faces used to calculate the Human Prior Constraint (HP) are $I_{Rs}$. As illustrated in Fig. 2 of the main paper, $I_{Rs}$ includes rendered faces under multiple predicted lighting conditions. We cannot apply perceptual loss to $I_{Rs}$ because we do not have corresponding ground truth images under the multiple decoupled lighting conditions. However, HP does not require such ground truths. Perceptual loss can indeed be applied between the final rendered result $I_{out}$ and input image $I_{in}$.
> Below, we present a comparison between the perceptual loss implementation and the HP we proposed. We can see that HP still has better performances, which can confirm it provides more effective constraints for the textures through rendered faces under multiple lighting conditions.
>
> |  | Perceptual Loss | HP(ours) |
> |--|--|--|
> |PSNR $\uparrow$ | 28.72 |**29.22** |
> |SSIM $\uparrow$ | 0.90 | **0.91** |
> |LPIPS $\downarrow$ | 6.46 | **6.36** |
>
> **Q3: The texture of D3DFR in Figure 3 is missing. In addition, the FFHQ-UV's texture is presented in UV space while NextFace and the proposed methods' texture are presented in image space, I suggest presenting them all in UV space or all in image space.**
>
> **A3:** Thank you for your suggestion. We agree that the texture presentation space needs to be consistent. To achieve this, we transform all UV-textures to the image space. This allows textures from D3DFR and CPEM to be visualized as colored faces, even if they do not have UV-textures. Fig. 16 and 17 in the rebuttal PDF are provided in this space. We will also update the texture images in the main paper accordingly in the revised version.

---

> > ### Comment · Reviewer_NbDK · 2024-08-12
> >
> > Thanks for your rebuttal, it addressed my main concerns. I will keep my score. Do you plan to release the fitting code? I think it is a great contribution to the community.

---

> > > ### Author Response · Authors · 2024-08-12
> > >
> > > Thank you very much for your response! We are encouraged that our rebuttal can address your concerns. We will release our source codes after we clean up them.

---

### Official Review · Reviewer_W5JR · 2024-07-11

**Soundness:** 2
**Presentation:** 3
**Contribution:** 3
**Rating:** 6
**Confidence:** 3

**Summary:**

This paper proposes a method to recover face albedo by disentangling the input image into an albedo and shading maps (called "light conditions"). The shading maps are generated by a network which are combined with the albedo texture (generated from AlbedoMM) that is rendered under a set of $n$ lighting conditions represented using spherical harmonics. The networks for predicting the shading maps, spherical harmonics and the texture are optimized through a photometric loss along with regularizations. In order to generate meaningful masks, a binarization constraint is added on the generated masks along with an area constraint. While the final renders are significantly better than prior work, two crucial comparisons are missing i.e against the Haar-wavelet basis (https://graphics.stanford.edu/papers/allfreqmat/) and the Albedo from AlbedoMM

**Strengths:**

1) From my perspective, the central contribution of this paper is a somewhat 'neural' spherical harmonics (SH) representation, where the 'neural' masks help model sharp illumination effects such as shadows which SH fails to capture. This is certainly an interesting direction to explore

2) The paper is well written

3) Quantitative and qualitative results are better than prior work (especially in albedo). However, some crucial comparisons are missing.

**Weaknesses:**

1) The authors have not compared against optimizing in haar-wavelets (https://graphics.stanford.edu/papers/allfreqmat/) which are designed specifically to model sharp illumination dependent effects. Without this comparison, it is hard to asses the improvements the proposed model offers over classical representations. I understand that such an optimization may be compute intensive, but it is necessary.

2) There are no results shown of the initial AlbedoMM texture the texture map is initialized from. AlbedoMM already gives a relatively uniform texture map that is free from lighting artifacts, it is unclear what additional benefits texture optimization yields.

**Questions:**

How many bands of SH are optimized? Did you investigate optimizing higher number of bands instead of generating the masks (again, as done in: http://graphics.stanford.edu/papers/allfreq/allfreq.pdf

**Limitations:**

Yes.

---

> ### Author Rebuttal · Authors · 2024-08-06
>
> **Q1: The authors have not compared against optimizing in haar-wavelets[1] which are designed specifically to model sharp illumination dependent effects.**
>
> [1] Triple Product Wavelet Integrals for All-Frequency Relighting
>
> **A1:** Thank you for your suggestion. Please note that our contribution lies not in proposing a universal neural illumination model for all scenes, but in decoupling lights to aid in recovering more accurate textures in 3D face reconstruction.
>
> Reference [1] offers a solution for modeling environment illumination using haar wavelets for relighting. However, [1] focuses on efficient rendering, with all geometrical and texture information pre-processed and fixed. This differs significantly from the 3D face reconstruction setting, where geometrical, texture, and illumination parameters are co-optimized, making illumination optimization more challenging.
>
> To our knowledge, haar-wavelet modeling of illumination has not yet been applied to 3D face reconstruction, and no open-source codes are available. To validate this idea, we create a baseline by replacing the environment map modeling in NextFace and NextFace* using Spherical Harmonics (SH) with haar wavelets.
>
> The optimization of wavelets is considerably slower than SH, making it difficult to conduct quantitative comparisons on the full datasets. Therefore, we conducted a comparison on a subset of 5 pairs of single images from VoxCeleb2. The quantitative comparisons are presented below:
>
> |  | NextFace | NextFace(haar) | NextFace* | NextFace*(haar) |Ours
> |--|--|--|--|--|--|
> |PSNR $\uparrow$ | 21.82 | 21.84 | 22.12 | 22.12 | **26.66** |
> |SSIM $\uparrow$ | 0.83| 0.84 | 0.84 | 0.84 | **0.90** |
> |LPIPS $\downarrow$ | 11.12| 11.37 | 10.51 | 11.30 | **6.29**|
>
> We observe that modeling global illumination with haar wavelets has limited improvements over the original SH on PSNR for our face reconstruction task. Although performance might be enhanced by modifying the wavelets or increasing the size of the environment map, such modifications are beyond the scope of this paper and would incur significant time costs. Optimizing such a 64 × 64 environment map from wavelets takes approximately 28 minutes on a 256 × 256 image, whereas our method takes about 340 seconds. Therefore, our method remains more efficient and effective at this time.
>
> **Q2: There are no results shown of the initial AlbedoMM texture the texture map is initialized from. AlbedoMM already gives a relatively uniform texture map that is free from lighting artifacts, it is unclear what additional benefits texture optimization yields.**
>
> **A2:** As demonstrated in Fig. 2 and Alg. 1 of the main paper, we use AlbedoMM to initialize the textures in Stage 2, while the textures are directly optimized in Stage 3. Therefore, the textures from Stage 2 are precisely the AlbedoMM textures.
>
> In Section A.4 of the appendix, on page 12, we have provided a comparison between the AlbedoMM textures (Stage 2) and the final textures (Stage 3). As shown in Fig. 10 of the appendix, the texture optimization adds more details, such as beards, to the AlbedoMM textures, resulting in a more realistic reconstruction.
>
> **Q3: How many bands of SH are optimized? Did you investigate optimizing higher number of bands instead of generating the masks again?**
>
> **A3:** In this work, we follow NextFace [1] by using 9-band spherical harmonics (SH) to model local illumination. Additionally, we provide quantitative results of our method modeled with a single global SH using 9, 12, 15, and 18 bands, by removing the $f(\cdot)$ and $g(\cdot)$.
>
> | B | 9 | 12 | 15|18|Ours
> |--|--|--|--|--|--|
> |PSNR $\uparrow$ |25.19 | 25.26| 25.27 | 25.34 |**29.22**
> |SSIM $\uparrow$ | 0.87 | 0.87 | 0.87 | 0.87 |**0.91**
> |LPIPS $\downarrow$ | 9.16 | 9.23 | 9.22 | 9.10 |**6.36**
>
> We observe that increasing the number of bands in a single global SH yields quite limited improvements. A possible reason is that the external occluded shadows on human faces represent drastic illumination changes in relatively small areas, which may not be appropriately modeled as a single global SH during optimization.

---

### Official Review · Reviewer_SyuP · 2024-07-12

**Soundness:** 3
**Presentation:** 2
**Contribution:** 3
**Rating:** 6
**Confidence:** 4

**Summary:**

This work tackles the problem of external shadows in single image face reconstruction. Specifically, when the input image contains foreign shadows, this often affects the quality of the estimated facial texture, as the external shadows often become baked into the texture or leave behind undesirable artifacts in the shadow region. The paper proposes a comprehensive solution to this problem, including a way to decouple the face image into the result of multiple lights as well as an Adaptive Condition Estimation (ACE) strategy to select which lights are present in the image. The paper further proposes multiple human face priors, such as a global prior to ensure the face texture hue is consistent with the initialization from a 3DMM model and a local prior to ensure that the smoothness of the face texture is similar to the initialization. Experiments demonstrate that the method is able to improve rendering performance on in-the-wild face images by rendering both source images with external shadows and target images without external shadows. The major improvements on target images without external shadows shows that the method is able to produce accurate facial textures that are minimally impacted by the presence of these shadows. Qualitative ablations are also provided to further aid in evaluating the method.

**Strengths:**

The improvement in rendering performance on target images (w/o external shadows) clearly shows that the method is able to minimize the impact of external shadows from the source image. The qualitative results further support this conclusion, as the facial textures do not contain external shadow artifacts.

Qualitative ablations are available to help assess the impact of each technical contribution in this work.

This work is sufficiently novel since it enables more accurate face reconstruction under the condition of external occlusions from the source image, which is a heavily understudied problem. It also goes beyond traditional work in this area since it considers the scenario that there could be more than one light in the image and proposes a method to estimate the set of lights illuminating the face.

**Weaknesses:**

For the ablation studies, it would be much more convincing to have tables with quantitative results demonstrating the margin of improvement of each component. Simply picking a few favorable qualitative examples is easy and not convincing.

In almost all examples in this work, the foreign (external) shadows involved are caused by hats. What about other types of foreign shadows caused by tree leaves, pens, paper, hands, etc? How are the resulting face textures in these situations? It would be nice to compare with the baselines on some images with more diverse foreign shadows. If it is difficult to find such images in the wild, you can find a small test set of 100 or so images from the paper "Portrait Shadow Manipulation (SIGGRAPH 2020)" with diverse foreign shadow shapes.

Some important citations are missing from this work, especially in the face relighting and portrait shadow removal domains. There are several face relighting methods that involve intrinsic decomposition of faces and illumination modeling, some of which involve ray tracing to handle self shadows:

1. SfSNet : Learning Shape, Reflectance and Illuminance of Faces in the Wild (CVPR 2018)
2. Face Relighting with Geometrically Consistent Shadows (CVPR 2022)
3. Learning Physics-guided Face Relighting under Directional Light (CVPR 2020)

In addition, the shadow removal domain is highly relevant to this work since a simple solution to this problem would be to perform a preprocessing step to remove the foreign (external) shadows from the image first using a portrait shadow removal method before performing face reconstruction. These methods should be cited and discussed in the paper and the authors should verify that their method outperforms this simple baseline:

1. Portrait Shadow Manipulation (SIGGRAPH 2020)
2. Unsupervised Portrait Shadow Removal via Generative Priors (MM 2021)
3. Blind Removal of Facial Foreign Shadows (BMVC 2022)

**Questions:**

Please see the experiments I suggested above, especially with regard to quantitative evaluations for ablations, evaluations on images with more diverse foreign (external) shadows, and comparing with the simple baseline of running a foreign shadow removal method on the images beforehand. I will be carefully reviewing the rebuttal as well as the opinions of the other reviewers to decide if I would like to change my rating. Please also factor in missing citations, as the two areas I mentioned are very relevant to this work.

**Limitations:**

The authors have at least made an attempt to include a limitations section, although more analysis on failure cases would be helpful. For example, does the method fail when the external shadow covers most of the face? There is no section on potential negative societal impact, although there are certainly some potential concerns with face reconstruction methods, as with any face modeling work. The potential to edit the face and produce fake content is always present.

---

> ### Author Rebuttal · Authors · 2024-08-06
>
> **Q1: Quantitative evaluations for ablations.**
>
> **A1:** Thank you for your suggestion. Here, we provide the quantitative ablation study of our proposed components below, following the same setting as Sec. 4.4 of the main paper. GP, LP, and HP denote our proposed constraints $L_{GP}$, $L_{LP}$, and $L_{HP}$ mentioned in Sec. 3.4, where Light and Occlusion denote the $f(\cdot)$ for combining multiple lighting conditions, and $g(\cdot)$ to remove the direct external occlusion, as claimed in Sec. 3.3, respectively.
>
> Here is the ablation study for constraints:
>
> |  | NA | GP | GP+LP|GP+LP+HP (ours)|
> |--|--|--|--|--|
> |PSNR $\uparrow$ |27.20|28.81| 28.82 |29.22 |
> |SSIM $\uparrow$ |0.88|0.89| 0.90 | 0.91 |
> |LPIPS $\downarrow$ |8.34|6.78| 6.49 | 6.36 |
>
> Here is the ablation study for the proposed $f(\cdot)$ and $g(\cdot)$.
>
> |  | NA | + Light ($f(\cdot)$) | + Occlusion ($g(\cdot)$)
> |--|--|--|--|
> |PSNR $\uparrow$ | 25.19|27.52 |29.22
> |SSIM $\uparrow$ |0.87 |0.89 | 0.91
> |LPIPS $\downarrow$ | 9.16|7.75 | 6.36
>
> We can see that each component has contribution to the final performances.
>
> **Q2: Missing citations and Discussions about relighting works [1, 2, 3].**
>
> [1]SfSNet : Learning Shape, Reflectance and Illuminance of Faces in the Wild (CVPR 2018)
>
> [2]Face Relighting with Geometrically Consistent Shadows (CVPR 2022)
>
> [3]Learning Physics-guided Face Relighting under Directional Light (CVPR 2020)
>
> **A2:** [1] proposes a framework for decomposing face attributes from in-the-wild images through physics-based rendering, while [3] predicts residuals to enhance diffuse relighting performance. [2] introduces ray-tracing to model geometrically consistent self-occlusion shadows for relighting. However, these methods primarily focus on self-occlusion shadows rather than external shadows. Due to character limitations, we will discuss these approaches in more detail in the revised version.
>
> **Q3: Evaluations on images with more diverse foreign (external) shadows.**
>
> [4] Portrait Shadow Manipulation (SIGGRAPH 2020)
>
> **A3:** Thank you for your suggestion, we introduce 15 pairs of data (30 images) from [4] for evaluation under more diverse external shadows. The quantitative results are presented below, where some qualitative results are presented in Fig. 16 of the rebuttal PDF. We can see that our method still outperforms other methods under faces with diverse shadows.
> |  |CPEM|D3DFR|NextFace| NextFace* | FFHQ-UV | Ours
> |--|--|--|--|--|--|--|
> |PSNR $\uparrow$ | 23.34| 25.02 |21.83| 21.93 | 24.00 | **28.97**
> |SSIM $\uparrow$ |0.84 |0.87 | 0.84  | 0.84 | 0.88 | **0.92**
> |LPIPS $\downarrow$ | 9.31|9.11 | 9.48 | 9.69 | 7.82 | **7.00**
>
> **Q4: Comparing with the simple baseline of running a foreign shadow removal method such as [4, 5, 6] on the images beforehand.**
>
> [5] Unsupervised Portrait Shadow Removal via Generative Priors (MM 2021)
>
> [6] Blind Removal of Facial Foreign Shadows (BMVC 2022)
>
> **A4:** It is an interesting idea. To validate its effectiveness, we introduce the shadow removal work [5] to pre-process the face images with shadows because it is the most recent shadow-removal work with a pre-trained model.
> Here are the quantitative results evaluated on data from [4].
> |  |CPEM|D3DFR|NextFace| NextFace* | FFHQ-UV | Ours
> |--|--|--|--|--|--|--|
> |PSNR $\uparrow$ | 25.56| 25.60 |23.49| 25.47 | 26.19 | **28.97**
> |SSIM $\uparrow$ |0.86 |0.88 | 0.86  | 0.87 | 0.90 | **0.92**
> |LPIPS $\downarrow$ |8.56|8.68 | 9.09 | 6.41 | 7.02 | **7.00**
>
> Here are quantitative the results evaluated on single images from VoxCeleb2.
>
> |  |CPEM|D3DFR|NextFace| NextFace* | FFHQ-UV | Ours
> |--|--|--|--|--|--|--|
> |PSNR $\uparrow$ | 24.84| 26.78 |23.77| 24.51 | 25.35 | **29.22**
> |SSIM $\uparrow$ |0.87 |0.90 | 0.85  | 0.86 | 0.91 | **0.91**
> |LPIPS $\downarrow$ |10.20|7.93 | 10.47 | 9.64 | 7.62 | **6.36**
>
> We also provide qualitative comparisons in Fig. 17 of the rebuttal PDF. The pre-processing with the shadow removal method indeed improves the performances of baselines, while our method still outperforms them.
>
> From the qualitative results, we observe that the shadow removal model cannot fully remove the external shadows in cases where the external shadows cover relatively large regions. Although modifying the shadow removal model to be more powerful may further improve the final performances, it goes beyond the range of this work. We can explore it in the future.
>
> **Q5: More analysis on failure cases. For example, does the method fail when the external shadow covers most of the face?**
>
> **A5:** As shown in Fig. 16 of the rebuttal PDF, our method performs well even under external shadows covering most of the face. We present visualizations of our inferior cases in Fig. 19 of the rebuttal PDF. As mentioned in the Limitations section of the main paper, our primary limitation is the initialization with AlbedoMM. For faces with many high-frequency details, such as wrinkles, our method may lose these details during reconstruction. Replacing AlbedoMM with more powerful face representations could address this issue. We plan to explore this further in future work.
>
> **Q6: There is no section on potential negative societal impact.**
>
> **A6:** Sorry for the lack of such discussion. Similar as existing human face reconstruction works, our method, focusing on recovering more realistic face textures under shadows from self and external occlusions, may also lead to the ethical concerns about privacy, consent, and the spread of misinformation. We will discuss about it carefully in the revised version.

---

> > ### Comment · Reviewer_SyuP · 2024-08-09
> >
> > Thank you for your rebuttal! Most of my concerns are addressed here. However, for evaluating in Q3, can we use all 100 images in that dataset? This would be more comprehensive and convincing to me than only evaluating on 15 of them.

---

> > > ### Author Response · Authors · 2024-08-10
> > > **Response to Reviewer SyuP**
> > >
> > > Thank you for your feedback. In the previous rebuttal, we presented results from 15 representative pairs due to time and hardware constraints. We have now extended our evaluation in Q3 to all 100 data pairs below.
> > >
> > > |  |CPEM|D3DFR|NextFace| NextFace* | FFHQ-UV | Ours
> > > |--|--|--|--|--|--|--|
> > > |PSNR $\uparrow$ | 23.31| 25.45 |22.16| 22.24 | 23.89 | **28.78**
> > > |SSIM $\uparrow$ |0.83 |0.86 | 0.84  | 0.84 | 0.88 | **0.91**
> > > |LPIPS $\downarrow$ | 9.93 |9.18 | 9.58 | 9.64 | 7.96 | **7.33**
> > >
> > > We can see that our method consistently outperforms the others, further validating its effectiveness.

---

> > > > ### Comment · Reviewer_SyuP · 2024-08-10
> > > >
> > > > Thanks for your response! I've raised my score to Weak Accept. Please remember to include these new comparisons in the final version to deliver a more convincing story as well as citing the relighting and shadow removal methods I mentioned previously to have a more comprehensive discussion of related methods.

---

> > > > > ### Author Response · Authors · 2024-08-10
> > > > > **Response for Reviewer SyuP**
> > > > >
> > > > > Thank you once again for your valuable time and constructive suggestions! We are inspired by your recognition of our method and our responses.  We will add all the mentioned comparisons and discussions to our final version.

---

### Official Review · Reviewer_WqKW · 2024-07-15

**Soundness:** 3
**Presentation:** 2
**Contribution:** 2
**Rating:** 6
**Confidence:** 4

**Summary:**

The paper presents a method for reconstructing 3D face textures from monocular images in the presence of occlusions, both self-occlusions and occlusions by other scene elements such as hats. The paper identifies a key limitation in existing works -- they all get impacted by the lighting changes introduced by an occluder, as they assume globally consistent lighting. The paper proposes to model the scene using a combination of different lighting for the different regions of the face. The different regions are reconstructed without direct supervision. Additionally, a mask is used to separate out the occluder pixels. Modeling multiple light sources enables the method to model appearance effects due to occluders and thus enables higher quality reconstructions than the state of the art.

**Strengths:**

The paper is very well motivated. It identifies a key problem and motivates the solution very well.
The method is novel and simple.
The results are impressive and outperform the chosen baselines. The reconstructed masks fit very well to the shadow boundaries

**Weaknesses:**

The main weakness is the lack of discussion and comparisons with existing works, and the lack of stronger baselines.

The paper does not mention existing research on dealing with occlusions. Here are a couple:
- Robust Model-based Face Reconstruction through Weakly-Supervised Outlier Segmentation [CVPR23]
- Occlusion-aware 3D Morphable Models and an Illumination Prior for Face Image Analysis [IJCV18]
Both these papers jointly perform 3D reconstruction and solve for occluders. They might not use texture optimization and instead perform per-vertex color optimization, but their techniques should be trivially extended for texture optimization. Comparisons to these baselines makes it impossible to evaluate this submission.

In addition, there are other strong baselines that were not considered. Generative models, such as those introduced by "Face De-occlusion using 3D Morphable Model and Generative Adversarial Network" [ICCV19] can remove occlusions from portrait images. Can the output of those methods directly be used with NextFace for high-quality reconstructions?

The other weakness relates to the exposition. The technical details of the paper were not easy to read. I did not understand how many number of masks are used, what 't' refers to in L128, how many such 't' are used. How are all the hyperparameters, such as 'n', n_L', m, epsilon, etc., chosen? I did not find a discussion. The paper is not reproducible in its current state.

**Questions:**

Please refer to weaknesses.

**Limitations:**

Yes

---

> ### Author Rebuttal · Authors · 2024-08-06
>
> **Q1: Lack of discussion and comparisons with 3D face reconstruction works dealing with de-occlusion, such as [1, 2].**
>
> [1] Robust Model-based Face Reconstruction through Weakly-Supervised Outlier Segmentation (CVPR'23)
>
> [2] Occlusion-aware 3D Morphable Models and an Illumination Prior for Face Image Analysis (IJCV'18)
>
> **A1:** Thank for your reminding. References [1] and [2] propose different methods to predict masks to remove external occlusions, where they optimize 3D Morphable Model(3DMM) according to the remained regions to reconstruct de-occluded faces. We will discuss these methods in more detail in the revised version.
>
> However, de-occlusion focuses on removing external occlusions, while **the shadows are not just occlusions**. A shadow region is actually the face under another illumination, which may include useful texture information. Simply treating shadowed regions as occlusions and removing them using de-occlusion methods may lead to the loss of face texture information in these areas. Our method, which integrates multiple lighting conditions, is more suitable for recovering textures from faces with shadows.
>
> Here, we present a comparison with the recent open-sourced de-occlusion method [1] by extracting its 3D textures and evaluating them using the texture quality metrics as mentioned in Section 4.1 of the main paper. The quantitative results, evaluated on single images from VoxCeleb2, are presented below.
> |  | Deocclu[1] | Ours |
> |--|--|--|
> |PSNR $\uparrow$ | 25.13 |**29.22** |
> |SSIM $\uparrow$ | 0.88 | **0.91** |
> |LPIPS $\downarrow$ | 14.09 | **6.36** |
>
> We also conduct comparisons on Video sequences from VoxCeleb2:
> |  | Deocclu[1] | Ours |
> |--|--|--|
> |PSNR $\uparrow$  | 25.03 |**29.15** |
> |SSIM $\uparrow$  | 0.88 | **0.91** |
> |LPIPS $\downarrow$ | 14.18 | **6.92** |
>
> Our method demonstrates superior performances. We also provide qualitative comparisons in Fig. 18 of the rebuttal PDF. Our method constructs more realistic results. For example, results from [1] miss details such as beards, while our method preserves them.
>
> **Q2: Can we use the de-occluded results of generative de-occlusion model such as [3] on NextFace for high quality reconstruction?**
>
> [3] Face De-occlusion using 3D Morphable Model and Generative Adversarial Network (ICCV'19)
>
> **A2:** Reference [3] trains a generative model to produce de-occluded face images from input and initial 3DMM synthesis images. We will discuss this in more detail in the revised version.
>
> However, [3] does not provide its source code, making a direct comparison unavailable at this time. Additionally, as stated in Sec. 5.1 of [3], the model is trained on synthetic data constructed by layering common non-face objects onto faces. This approach focuses on learning how to remove these objects, without any consideration about shadows.
>
> As we mainly discuss about **the influence of shadows from self and external occlusions** in this work, a closer idea is using 2D shadow removal methods mentioned by **Q3-Reviewer SyuP**. Such works desire to introduce generative model to recover high quality images free from shadows.
>
> We conducted experiments using the output of the shadow removal method [4] as input to CPEM, D3DFR, NextFace, NextFace*, and FFHQ-UV to evaluate their ability to acquire accurate textures without shadows. The quantitative comparisons are presented in **Q3-Reviewer SyuP**, and we also provide qualitative comparisons in Fig. 17 of the rebuttal PDF.
>
> Our results show that introducing the output of such generative models to the texture modeling baselines still yields inferior results compared to our method.
>
> [4] Unsupervised Portrait Shadow Removal via Generative Priors (MM 2021)
>
> **Q3: The technical details of the paper were not easy to read. I did not understand how many number of masks are used, what 't' refers to in L128, how many such 't' are used. How are all the hyper-parameters, such as 'n', n_L', m, epsilon, etc., chosen?**
>
> **A3:** Sorry about the confusing parts. The number of masks are actually changing according to the illumination complexity of the image. As illustrated in Line 117~124 of the paper, we initialize $n$ separate lighting conditions, where $f(\cdot)$ predicts one mask for each Lighting condition. The $n$ masks make up $M_N$ in Fig.2. Then, the parameters of $n$ lighting conditions and $f(\cdot)$ are optimized to construct the face image, where some lighting conditions not existing in the face image will get smaller mask areas as illustrated in Fig.2 of the paper.
>
> Then, in ACE, we drop such lighting conditions and masks with small mask areas, as explained in Line 142~147 of the paper. In this way, $n_L$ masks in $M_N$ are remained, which make up $M_L$. We can see that $n_L$ is a adaptive number less than $n$.
>
> We select $n=5$ in this work. It means there would be 5 masks in $M_N$ at the beginning. Where $n_L<=n$ would be decided according to the subsequent optimization. Corresponding discussions could be found in **Q1-Reviewer NbDK**.
>
> $m$ mentioned in Line 142 to 157 is not a hyper-parameter, it represents the mask value between 0 ~ 1 at a pixel position, e.g., $m=0$ at the black regions of the masks.
> $\epsilon$ is a threshold to filter out the masks with quite small areas. We set it as 0.17 in this work.
>
> $t$ is decided by the number of frames in the input image/video sequence. For the single image reconstruction, $t$ is set as a fixed 0 value, where it would be $i/k$ for the $i_{th}$ frame of a $k$ frames video sequences. In this work, we mainly conduct comparisons on 8-frame sequences as claimed in Line 203, page 7 of the paper.
> We will add more details about the mentioned contents in the revised version.

---

> > ### Comment · Reviewer_WqKW · 2024-08-11
> >
> > Thanks for the response, and for clarifying that some papers I mentioned were focussed only on occlusions rather than shadows. However, Robust Model-based Face Reconstruction through Weakly-Supervised Outlier Segmentation [CVPR23] deals with outliers generally, including shadows? The qualitative results in the pdf seem to show that their results are indeed robust to shadows. Their results do not capture details such as beards very well, but that is an orthogonal issue to robustness to shadows? I suspect these details can be captured by lowering the weight of the regularizer in Eq.8 of their paper (https://arxiv.org/pdf/2106.09614). Can you comment on this?

---

> ### Author Response · Authors · 2024-08-12
>
> Thank you for your response! We apologize for the confusion regarding the qualitative results.
> We agree that the work "[1] Robust Model-based Face Reconstruction through Weakly-Supervised Outlier Segmentation [CVPR23]" can avoid shadow effects on the textures. But **please note that it is different from our method and limited by texture details.** [1] predicts a binary mask to remove these shadow regions, while our method reconstructs them with different illuminations.
>
> The details, e.g. beards, may be difficult to be captured by [1] because it fully relies on a **pure linear parametric 3D Morphable Model (3DMM)** to model the face textures and distinguish the occlusion regions. As claimed in Sec.3.1 of [1], it models face textures with 3DMM parameters predicted by networks.
> Although 3DMM does not have explicit shadows on its textures as shown in the rebuttal PDF, it is hard to construct details, e.g. mentioned beards, as claimed in “Parametric Face Models”, Sec. 2 of [2]. Therefore, even lowering the weight of the regularizer in its Eq.8 may not be able to capture the details due to the limitation of the 3DMM model itself. It is also mentioned in the Limitation, Sec. 4.5 of [1] that a face model capable of depicting facial details, makeup, and beard is required. While incorporating such a model could potentially address this issue, doing so requires substantial effort and falls outside the scope of our paper, which we leave for future work.
>
> [2] Self-supervised Multi-level Face Model Learning for Monocular Reconstruction at over 250 Hz
>
> In our method, although we also initialize face textures using a parametric AlbedoMM in Stage 2, our direct texture optimization in Stage 3 allows us to add fine details to the textures, as shown in Figure 10 of the appendix in our paper. This can be achieved because we model the shadows with illumination to avoid their influence on textures. Directly optimizing the textures in [1] may not achieve this. As explained in Sec. 3.2 of [1], it treats regions which cannot be well described with 3DMM under a global illumination, e.g. the beard regions, as occlusions and learns to predict a mask to remove them. Since these regions are removed, they cannot be further used to refine the 3DMM textures through optimization.
>
> Since we are not allowed to include more qualitative results in this comment, we will provide additional visualizations of the predicted masks from [1] in the revised version to support our claim. Furthermore, the quantitative comparisons in Q1 also confirm the effectiveness of our method, which will be discussed in greater detail in the revised version.
>
> Thank you once again for your feedback. Please feel free to reach out if you have any further questions.

---

> > ### Comment · Reviewer_WqKW · 2024-08-13
> >
> > Thanks, I now understand the distinction between ignoring the outliers in [1] and modeling them in this submission. I will raise my score.

---

> > > ### Author Response · Authors · 2024-08-13
> > >
> > > Thank you for your comment! We are happy that our response can address your problem. We will add the former mentioned discussions to the revised version.

---

### Author Rebuttal · Authors · 2024-08-06

We thank all reviewers for your valuable comments. We are inspired that all reviewers recognize the sufficient motivation and good performance of our method. We apologize for any confusion caused by missing comparisons, citations, unclear definitions, and other issues. In this rebuttal, we make reply to each concern carefully, point by point.
All quantitative comparisons are based on the texture quality metric for target images as defined in Line 198-203 of the main paper. We also present qualitative results in the attached rebuttal PDF. We hope our response addresses your concerns well. Please feel free to comment if you have any further questions or suggestions about this work.

---

### Decision · Program_Chairs · 2024-09-25

**Decision:**

Accept (poster)

**Comment:**

After the rebuttal and extensive reviewer-author discussion, all four reviewers proposed acceptance, thus, the AC recommends acceptance. The AC strongly encourages the authors to include the missing comparisons, citations, and the rest of the rebuttal/discussion material in the final version of the paper.